# OpenHA: A Series of Open-Source Hierarchical Agentic Models in Minecraft

Zihao Wang [* 1]  Muyao Li [* 1]  Kaichen He [* 1]  Xiangyu Wang [1]  Zhancun Mu [1]  Minghao Liu [1]  Anji Liu [2]  Yitao Liang [1]

## Abstract

A critical challenge in developing capable AI agents is defining their "action space"—the set of possible actions they can take. These spaces can range widely, from generating code and using language skills to operating on latent representations or raw joystick controls. Through a large-scale study in Minecraft, we discovered a major dilemma: no single action space is universally best. The most effective action space is highly task-dependent, which complicates the goal of building one generalist agent that can handle everything. To solve this, we introduce Chain-of-Action (CoA), a novel framework that unifies high-level abstracted actions and low-level control actions within a single model. With CoA, an abstract goal is not just a final command; instead, it serves as an intermediate reasoning step that guides the model to generate the precise, executable actions needed to complete the task. Furthermore, we show that a generalist agent, trained on a diverse mix of action spaces using CoA, learns a more generalizable policy. This unified agent achieves a new state-of-the-art, outperforming strong, specialized baselines. To support the research community, we are releasing the OpenHA (Open Hierarchical Agents) suite, which includes our benchmark of over 800 tasks, curated datasets, source code, and all model checkpoints at: `https://github.com/CraftJarvis/OpenHA`.

## 1. Introduction

In their early designs, AI agents were primarily built as modular systems that utilized pre-trained models, including Large Language Models (LLMs) (Touvron et al., 2023; OpenAI, 2023a; Yang et al., 2025; Anthropic, 2025) and Vision-Language Models (VLMs) (Achiam et al., 2023; Wang et al., 2024b; Team et al., 2023; Guo et al., 2025) and combined with external modules—such as memory, planning, and tool use—to generate actions within an environment (Weng, 2023; Yang et al., 2023; Cheng et al., 2024). However, this modular approach leads to a significant limitation: the agent's performance is restricted by the capabilities of the pre-trained model (Wang et al., 2024a; Yu et al., 2025) and is unable to learn and improve from its interactions with the environment (Li et al., 2025b).

To address this constraint, recent advancements have turned toward the development of end-to-end pre-trained agentic models that learn directly from interaction data (Qin et al., 2025; openai, 2025; Brohan et al., 2023b; Driess et al., 2023). These models embed the capacity for action generation within the agent itself. A key distinction among different agentic models lies in how they represent the action space of agents, which ranges from language skill (Driess et al., 2023), code (Wang et al., 2024a; Liang et al., 2022), MCP (Anthropic, 2024), GUI actions(within computer os) (openai, 2025), and raw actions(within robotics) (Brohan et al., 2023b). Based on the hierarchy of actions, these models can be broadly categorized into two types. The first type, Vision-Language-Action (VLA) models (Kim et al., 2024; Li et al., 2025a), directly tokenize low-level actions—such as keyboard inputs or robotic movements—into sequences that are compatible with the vocabulary of language models. This enables the model to predict and generate low-level actions directly from the interaction context. However, a significant challenge arises from the semantic gap between the high-level abstractions of language and the continuous, detailed nature of low-level actions (Wang et al., 2024d).

To overcome this issue, a second class of agent models adopts a hierarchical architecture and defines an abstracted action space for agentic models (Belkhale et al., 2024b; Driess et al., 2023; Bjorck et al., 2025). In these models, a high-level language model predicts an abstracted action,

---
[*]Equal contribution  [1]Institute of Artificial Intelligence, Peking University, Beijing, China [2]School of Computing, National University of Singapore, Singapore. Correspondence to: Zihao Wang <zhwang@stu.pku.edu.cn>, Muyao Li <2350076251@qq.com>, Anji Liu <anjiliu@nus.edu.sg>, Yitao Liang <yitaol@pku.edu.cn>.

*Proceedings of the 43rd International Conference on Machine Learning*, Seoul, South Korea. PMLR 306, 2026. Copyright 2026 by the author(s).

which is then decoded into a sequence of low-level actions by a learned action tokenizer or controller (Wang et al., 2024d; Cai et al., 2024b). While this hierarchical approach separates reasoning and control, a key question arises: *how can abstracted action spaces be effectively defined and formulated from the continuous stream of low-level actions?* Despite the widespread adoption of various action tokenizers (Zhong et al., 2025a), a clear, standardized evaluation remains lacking. Most action tokenizers are tested on domain-specific benchmarks and proprietary datasets, making direct and fair comparisons difficult (Belkhale et al., 2024a; Gu et al., 2023; Kim et al., 2024).

We present a unified and fair evaluation to determine which abstracted action spaces are most effective. This evaluation includes a large-scale comparison of abstracted action spaces across 800 tasks, based on the open-ended environment of Minecraft (Fan et al., 2022). Our experiments reveal that the effectiveness of an action tokenizer is highly task-dependent, with no single action space demonstrating universal superiority.

Traditional abstracted actions are limited and only applied to hierarchical agentic models, which include high-level vision-language models and learning-based decoders or action tokenizers. In these models, joint optimization across different levels is challenging (Driess et al., 2025). We further explored whether these abstracted actions are also beneficial for the end-to-end VLA model. We introduce the Chain-of-action (CoA) framework, which integrates the strengths of both hierarchical agentic models and VLA models. In CoA, action generation is structured as a two-step, auto-regressive process: first, the model generates a high-level abstracted action, which serves as an intermediate "thinking"; then, conditioned on this thought, the model generates the low-level action auto-regressively.

In addition, inspired by the discovery that different abstracted actions excel in distinct task domains, we propose an All-in-One training strategy. The key insight is that by grounding the model's final output in low-level actions, CoA enables a unified prediction structure across all action spaces. This allows us to train a single agent on a composite dataset containing multiple action representations. The resulting agent can master a diverse range of actions, outperforming specialist models trained on specialized actions.

The main contributions of this work are fourfold: 1) We provide a large-scale and systematic evaluation to demonstrate that the optimal abstracted action space is task-dependent. 2) We introduce the Chain-of-Action framework to synergize high-level reasoning and low-level control by using abstracted actions as intermediate thinking. This framework bridges the gaps and offers superior performance. 3) We demonstrate that training a single agentic model on mixed datasets of diverse action spaces enhances its

decision-making ability to generalize across tasks. 4) We release a comprehensive suite including trained checkpoints, code, and datasets for various agentic models to support further research in action spaces and generalization.

## 2. Abstracted Actions in Agentic Models

### 2.1. Hierarchical Agent Structure

Hierarchical agents model the agent's decision-making process through a hierarchical policy decomposition. This process consists of two levels: first, a high-level pretrained LLM-based agentic model generates an abstracted action based on the task instruction and the current observation; second, a low-level policy acts as the **action tokenizer** and uses this abstracted action as guidance to produce a primitive, executable action in the environment. This two-level procedure can be expressed as:

$$A \sim \pi_{LM}(\cdot \mid \text{obs}, \text{ins}), a \sim \pi_{policy}(\cdot \mid \text{obs}, A) \quad (1)$$

Here, $ins$ denotes the human-provided textual instruction, and $obs \in \mathbb{R}^{H \times W \times 3}$ represents the current visual observation. The variable $A \in \mathcal{A}_{abs}$ signifies an **abstracted action** sampled from a high-level action space $\mathcal{A}_{abs}$, generated by the high-level policy $\pi_{LM}$. Subsequently, $a \in \mathcal{A}_{env}$ represents a **primitive environmental action** from the executable action space $\mathcal{A}_{env}$, generated by the low-level policy $\pi_{policy}$ as the action tokenizer (Zhong et al., 2025a). In domains like computer interaction, these primitive actions correspond to raw inputs such as keyboard presses or mouse movements.

The specific formulation of the abstracted action $A$ is a critical design choice and varies significantly across different agent architectures. It can range from symbolic, language-based skills (e.g., "pick up the apple") (Driess et al., 2023), to continuous, grounded interaction trajectories (Gu et al., 2023), textual descriptions of motion (e.g., "go forward") (Belkhale et al., 2024b), discrete latent codes learned by encode-decoder models (Wang et al., 2024d; Bjorck et al., 2025), or even future visual tokens (Zhou et al., 2024; Zhao et al., 2025).

In typical implementations, the high-level auto-regressive models $\pi_{LM}$ are instantiated as a large-scale Vision-Language Model (VLM) (Wang et al., 2024b; Grattafiori et al., 2024; OpenAI, 2023b), leveraging its powerful reasoning capabilities to process multimodal inputs and generate the abstracted actions $A$. In contrast, the low-level policy $\pi_{policy}$ is often a more lightweight, specialized model, such as a Transformer with less than 100M parameters (Dai et al., 2019; Vaswani et al., 2017), tasked with translating the plan $A$ into a sequence of environment-specific primitive actions $a$ (Lifshitz et al., 2024; Cai et al., 2024b).

This hierarchical formulation is general enough to subsume many recent Vision-Language-Action (VLA) models (Chi

*Table 1.* Definitions and examples of the different action spaces in agents.

| Symbol | Definition | Action Types | Example |
|---|---|---|---|
| $a$ | Env Action | 24 | {"camera": [-1, -9], "ESC": 0, "back": 0, "drop": 0, "forward": 0, "hotbar.1-9": 0, "inventory": 0, "jump": 0, "left": 0, "right": 0, "sneak": 0, "sprint": 0, "swapHands": 0, "attack": 0, "use": 0, "pickItem": 0} |
| $a_{raw}$ | Raw Action | 35 | "<\|reserved_token_1\|><\|reserved_token_7\|><\|reserved_token_9\|><\|reserved_token_2\|>" |
| $a_{text}$ | Text Action | 3 | "Action: move(0, 0) and press(left.control, w, a) and click(left)" "Action: move(4, -1) and press(left.control, w))" |
| $a_{mot}$ | Motion Action | - | "Motion: Go forward, Turn left." |
| $a_{ground}$ | Grounding Action | 8 | "Grounding: Mine(object='oak_log', position=[100, 200])" "Grounding: Approach(object='sheep', position=[200, 300])" |
| $a_{lat}$ | Latent Action | 15000 | "<\|reserved_token_2\|>" |

et al., 2023; Team et al., 2024; Brohan et al., 2022; 2023a; Zheng et al., 2024; Zhang et al., 2024; Zhong et al., 2025b; Li et al., 2024; Zhang et al., 2025; Zhu et al., 2025; Zhou et al., 2025; Chen et al., 2025). These non-hierarchical or "flat" VLA models can be viewed as a special case of our framework where the hierarchy collapses and the policy $\pi_{policy}$ is the language tokenizer itself (Bai et al., 2023). In this scenario, the high-level policy is trained to directly output the specification of the primitive action, effectively making the abstracted action space equivalent to the primitive one ($A \equiv a$). Consequently, the low-level policy $\pi_{policy}$ reduces to a trivial identity function, as the planning and execution steps are merged into a single end-to-end network.

## 2.2. Action Spaces in Hierarchical Agents

The efficacy of a hierarchical agent is intrinsically tied to the design of its action space, $A$. This space defines the vocabulary of behaviors the agent can command (Zhong et al., 2025a). In this work, we investigate a spectrum of action spaces, ranging from low-level environmental controls to various high-level abstractions. A summary and examples of these spaces are provided in Table 1 and in Appendix A.

**Primitive Environmental Actions** ($a \in \mathcal{A}$)**.** At the most fundamental level is the primitive action space dictated by the environment simulator. In our context, this corresponds to low-level keyboard and mouse operations. To be processed by a language-based model, these raw actions must be tokenized using reserved tokens within the language model $a_{raw} \in \mathcal{A}_{raw}$ like OpenVLA (Kim et al., 2024; Brohan et al., 2023a) or textual serialization as text action $a_{text} \in \mathcal{A}_{text}$ mouseMove(dx=4, dy=-1) (Seed, 2025; Team, 2025; Wang et al., 2025).

**Language Skills** $a_{skill} \in \mathcal{A}_{skill}$**.** A natural choice for high-level models based on VLMs is to use natural language skills as the abstracted action space. These skills typically describe a complete, goal-oriented behavior, often with a verb-object structure like "chop down trees" or "open the chest". Executing such a skill requires a sophisticated, language-conditioned low-level policy that interprets the

command and, conditioned on the current visual observation, decodes it into a sequence of primitive actions to achieve the goal (Wang et al., 2023; Driess et al., 2023).

**Motion Primitives** $a_{mot} \in \mathcal{A}_{mot}$**.** Occupying an intermediate level of abstraction, motion primitives represent temporally extended but object-agnostic movements. For instance, a continuous sequence of "press W and Left Shift" can be abstracted into a single "sprint forward" command, and a series of mouse movements can be categorized as "turn left". These actions are more abstract than primitive environmental actions but are lower-level than language skills, as they do not specify an object of interaction (Lifshitz et al., 2024; Belkhale et al., 2024b).

**Grounding Actions** $a_{ground} \in \mathcal{A}_{ground}$**.** While pure language skills face scalability challenges, as the low-level policy must learn to interpret possible object for a given skill. Grounding actions address this by creating a structured, multimodal representation. They retain the symbolic verb from language skills (e.g., mine) but replace the object's name with its grounded spatial coordinates in the visual observation (e.g., point=[100, 200]). This decouples the "what" from the "where," enabling a single, grounding-conditioned policy to execute a skill on any object by simply targeting its location, significantly improving generalization (Zhong et al., 2025b; Cai et al., 2024a; Lee et al., 2025).

**Latent Actions** $a_{lat} \in \mathcal{A}_{lat}$**.** The aforementioned action spaces rely on human-defined heuristics and feature engineering. In contrast, latent actions are learned directly from data in a self-supervised manner. This is typically achieved using an encoder-decoder framework, where an encoder compresses sequences of primitive actions into a continuous latent embedding (Van Den Oord et al., 2017). The high-level policy then generates these learned latent tokens as its abstracted actions (Deng et al., 2025; Yuan et al., 2024; Shafiullah et al., 2022).

**Other Abstracted Actions.** In addition to the aforementioned actions, recent works also explore other types of abstracted actions, such as future visual images $\mathcal{A}_{image}$ (Zhou et al., 2024; Zhao et al., 2025), rendered trajectories $\mathcal{A}_{traj}$

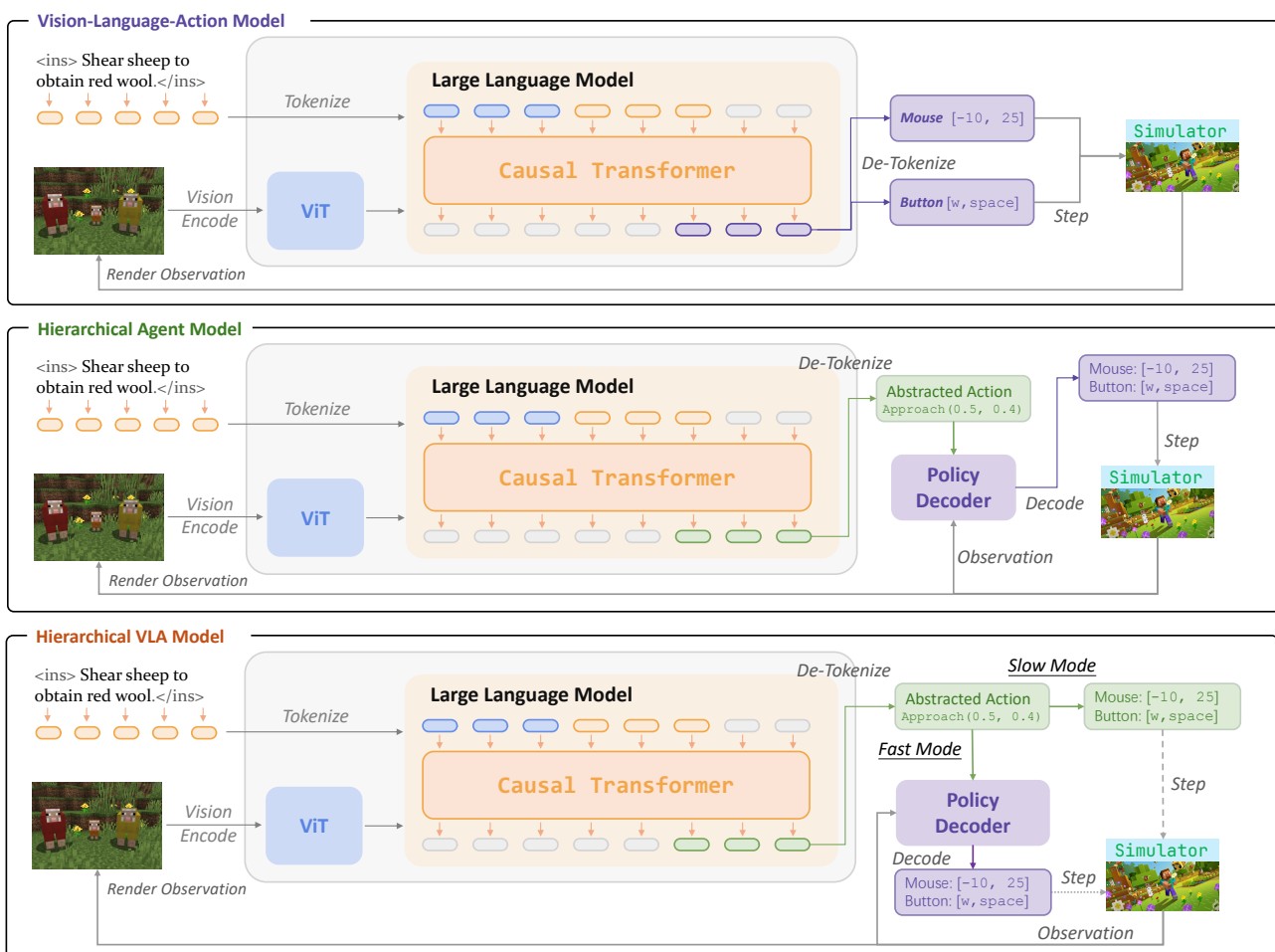

*Figure 1.* Comparisons between end-to-end vision-language-action (VLA) models and hierarchical agent (HA) models. The core difference between HA and the VLA model is that HA uses a policy-based action tokenizer to decode actions, while the vision-language-action model reuses the text tokenizer to decode actions. We further propose integrating the abstracted actions of the hierarchical agent into the training and inference processes of VLA, resulting in the Hierarchical VLA model, which has both fast and slow inference modes, using a policy-based action tokenizer and a text tokenizer to decode environmental actions, respectively.

(key-points and lines) (Zheng et al., 2024; Lee et al., 2025; Huang et al., 2024; 2023) and executed codes $\mathcal{A}_{\text{code}}$ (Liang et al., 2022; Singh et al., 2022; Wang et al., 2024a).

# 3. Synergizing Abstracted Decisions and Low-level Control

## 3.1. Taking Abstracted Actions as Thoughts

While abstracted actions simplify the decision-making for VLM-based agents, they are usually tied to two-stage hierarchical architectures. Such setups always hinder end-to-end training (Wang et al., 2024c; Driess et al., 2023). The low-level policy is learned in isolation, and often caps overall performance. To circumvent these limitations, we introduce the **Chain of Action (CoA)**, an auto-regressive formulation that reframes action generation as a single, sequential process. The core idea is to treat the high-level abstracted

action $A$ not as a command for a separate module, but as an intermediate reasoning step that guides the subsequent prediction of the low-level environmental action $a$.

This approach leverages the powerful sequence modeling capabilities of modern auto-regressive transformers (Zawalski et al., 2024; Wei et al., 2022; Vaswani et al., 2017). Instead of predicting the final action $a$ directly from the observation $obs$ and instruction $ins$, the model first generates the abstracted decisions $A$ and then, conditioned on this self-generated context, predicts the primitive action $a$. We formalize this by modeling the joint probability of the action chain $(A, a)$. Given an instruction ins and an observation obs, the likelihood of generating the sequence is factorized

using the chain rule of probability:

$$P(A, a \mid \text{ins}, \text{obs}) = \underbrace{P(a \mid \text{ins}, \text{obs}, A)}_{\text{Action Generation}} \cdot \underbrace{P(A \mid \text{ins}, \text{obs})}_{\text{"Thought" Generation}}$$

(2)

Here, a single auto-regressive language model, $\pi_\theta$, models the entire generation process. The entire model $\pi_\theta$ is trained end-to-end by maximizing the log-likelihood of this joint probability over a dataset of expert trajectories.

To illustrate, consider the task "CHOP DOWN A TREE." A standard VLA model must implicitly reason about the tree's location, its distance, and the required navigation, all while directly outputting a complex sequence of low-level actions (e.g., `press W`, `turn left`, `left click mouse`). This implicit reasoning is difficult to learn. In contrast, the CoA agent explicitly decomposes this problem:

1. **Thought Generation**: The model first generates a grounded action as a thought, such as $a_{\text{ground}}$: `Attack(object=oak_log, point=[100, 200])`.

2. **Action Generation**: Conditioned on this explicit target, the subsequent prediction of low-level actions becomes a simpler, more grounded task. The model can observe that the target at `[100, 200]` is far away and to the left, making the generation of $a$: `press W` a direct consequence of the explicit reasoning step.

In this way, the CoA formulation grounds complex reasoning in an explicit, intermediate representation, breaking down a difficult prediction problem into a more manageable, sequential one and seamlessly unifying hierarchical reasoning within a single end-to-end model.

### 3.2. Hierarchical Agentic Models with Fast or Slow Inference Methods

The Chain-of-Action (CoA) formulation is not only a training paradigm but also an architectural principle that affords significant flexibility at inference time. As illustrated in Figure 1, our model can operate in two distinct inference modes, allowing for a dynamic trade-off between reasoning depth and computational efficiency.

**Decoupled Inference Mode (Fast).** This mode emulates the classic hierarchical agent architecture. The primary large language model, which we refer to as the high-level policy $\pi_{LM}$, is queried only to generate the abstracted action $A$. The subsequent task of decoding it into a sequence of primitive actions $\{a^1, \ldots, a^k\}$ is offloaded to a separate, lightweight low-level policy $\pi_{policy}$ as a de-tokenizer. This approach significantly reduces the inference load on the large VLM, as it is invoked only once per high-level abstracted action, while the more efficient $\pi_{policy}$ can interact

with the environment for multiple steps. However, this mode has two primary drawbacks. First, it typically requires a multi-stage training process: the low-level policy $\pi_{policy}$ must be trained first to serve as a decoder, followed by the training of the high-level policy $\pi_{LM}$. Second, the overall performance is potentially bottlenecked by the capability of the smaller, specialized low-level policy.

**Unified Autoregressive Mode (Slow).** In contrast, this mode fully leverages the end-to-end nature of the CoA framework. The single, unified model $\pi_\theta$ is responsible for generating the entire action sequence autoregressively, first producing the abstracted action $A$, and then immediately generating the corresponding low-level action $a$, as described in Equation 2. This approach ensures that the full reasoning capacity of the large model is applied to both high-level thinking and low-level control, eliminating any potential performance bottlenecks from a separate decoder. The primary trade-off is the increased computational cost at inference time, as the VLM must generate a longer sequence of tokens for each environmental action, resulting in slower decision-making.

A crucial advantage of our framework is that these two inference modes do not require separately trained models. A single agent can be endowed with both capabilities through a unified, end-to-end training process on a mixed dataset. During training, we provide the model with examples formatted for both objectives. The final output format is controlled via a specific system prompt provided as input to the model. This methodology effectively merges the traditional hierarchical agentic models and the end-to-end VLA paradigm into a single, versatile architecture. It empowers a practitioner to dynamically select the most suitable inference cost at runtime—choosing the fast, decoupled mode for simpler tasks or resource-constrained environments, and the slower, more powerful unified mode for tasks demanding complex reasoning.

### 3.3. OpenHA: Training One Generalist Agent to Master Diverse Abstracted Actions

The traditional agentic model is usually trained on a fixed, predefined action space. We propose the **All-in-One** training strategy, which aims to produce a single generalist agentic model proficient in multiple action spaces. A naive approach would be to simply train a model on a heterogeneous dataset of trajectories, $\mathcal{D} = \bigcup_k \mathcal{D}_k$, where each sub-dataset $\mathcal{D}_k$ contains trajectories annotated with a different high-level action space $\mathcal{A}^{(k)}$ (e.g., Grounding Actions $GA$, Motion Actions $MA$, etc.). However, such a strategy faces a critical challenge: the different action spaces are fundamentally disparate. Without a shared basis, the model may treat them as isolated prediction domains, failing to generalize or transfer knowledge between them.

*Table 2.* Training recipes for different agent models.

| Model | HA | VLA | Hierarchical VLA | | OpenHA Model | |
|---|---|---|---|---|---|---|
| Stage | 1 | 1 | 1 | 2 | 1 | 2 |
| Datasets | $D^A[GA|MA|SA]$ | $D^a$ | $D^A+D^a$ | $D^{CoA}$ | $D^A[MA|GA]+D^a$ | $D^{CoA}[MA+GA]$ |
| Consumed Tokens | 1.58B | 3.59B | 3.33B | 0.27B | 3.40B | 0.22B |
| Learning Rate | 5e-6 | 1e-5 | 1e-5 | 5e-6 | 1e-5 | 5e-6 |
| Training Steps | 4000 | 10400 | 10800 | 400 | 8200 | 300 |

Our CoA framework provides a principled solution to this challenge. By adopting the CoA data format for all trajectories—$(ins, \{(o_t, A_t, a_t)\}_{t=1}^T)$—we ensure that every high-level abstracted action $a$, regardless of its origin space, is ultimately grounded in the same primitive environmental action space $\mathcal{A}$. The primitive action $a_t$ acts as a **common currency**, creating a shared semantic ground that connects all high-level abstractions.

This common grounding enables the model to learn the underlying relationships and functional equivalences between different action representations. For example, the model can infer that a high-level thinking expressed as a Grounding Action like $a_{\text{ground}}$: `Approach(object=sheep)` and one expressed as a Motion Action like $a_{\text{mot}}$: `Go forward` can be contextually similar if they both result in the same primitive action sequence $a$: `press(w)`. The training objective remains identical to that defined in Equation 2, maximized over the full, heterogeneous dataset $\mathcal{D}$.

We hypothesize that this All-in-One training strategy will yield two significant benefits. First, it will produce a highly versatile agent capable of understanding and generating multiple types of action abstractions. Second, and more importantly, by learning a more holistic and interconnected representation of behavior, the All-in-One generalist agent will develop a more robust and generalizable internal policy, ultimately outperforming any specialist agent trained on only a single form of action abstraction.

## 4. Experiments and Insights

In this section, we present a comprehensive suite of experiments designed to empirically validate our proposed frameworks and investigate key questions surrounding action representation for open-ended agents. We begin by detailing our experimental setup, including the training protocols for our models, the large-scale benchmark constructed for evaluation, and the metrics used to measure performance. Our analysis is then structured to systematically answer three central research questions: **Q1**: Within a traditional Hierarchical Agentic model (HA) paradigm, what type of abstracted action space is most effective in a complex, open-ended environment like Minecraft? **Q2**: Can incorporating these abstractions as intermediate reasoning steps within a

unified VLA model improve its overall performance? **Q3**: Does training a single agent on a heterogeneous mixture of action spaces enable it to learn a more generalizable and effective decision-making policy?

### 4.1. Experimental Setup

**Simulator and Evaluation Benchmarks.** We employ Minecraft as our primary testbed (Guss et al., 2019). Previous work on Minecraft agents has often been evaluated on a limited number of canonical tasks (Baker et al., 2022; Lifshitz et al., 2024; Cai et al., 2024b). Although MCU (Lin et al., 2023) and Minedojo (Fan et al., 2022) both claim to have around 3,000 tasks, these are usually automatically generated by LLMs, and cannot guarantee a fair and stable success rate for the agents. To enable a more comprehensive and fair comparison of different action spaces, we construct a large-scale benchmark comprising above 800 distinct tasks. All tasks are designed and verified by hand Appendix G. We categorize these tasks into three groups based on the primary capabilities required: 1) **Embodied** tasks requiring navigation and physical interaction within the 3D world (e.g., finding and chopping down a specific type of tree), 2) **GUI** tasks that involve complex interactions with graphical user interfaces, such as crafting items at a crafting table or smelting ores in a furnace, 3) **Combat** tasks that require the agent to engage with hostile mobs, focusing on survival and combat skills. To rigorously test for generalization, we ensure that all evaluation environments are out-of-distribution with respect to the training data. This is achieved by sampling novel initial world seeds and spawn locations for each task evaluation from Minecraft's vast procedural generation space. Each agent is evaluated on every task in our benchmark for a minimum of three independent runs, each with a different world seed. Our primary metrics are average **Success Rate** and **Steps to Completion**.

**Dataset Curation.** The training data for all agents are derived from the same expert dataset (Baker et al., 2022). On top of the raw data, we developed a **programmatic labeling pipeline** to facilitate the training of agents with high-level action spaces. This process yields three distinct types of trajectory datasets used in our experiments: (1) **High-Level Action Data** ($D_A$)**:** Trajectories containing only observations and their corresponding high-level ab-

*Table 3.* Evaluation results of Minecraft agents on over 800+ tasks. For each task category, we report the average steps to finish tasks, the best-of-10 (bo10) success rates, and the average success rate (ASR) across all tasks in that category (with $\pm$ standard deviation). Results highlighted in blue represent the second-best performances, while those in red indicate the state-of-the-art performance for each metric across all agents. '-' signifies tasks where the agent failed to achieve any success.

| Model | Action Tokenizer | Inference | Embodied Tasks | | | GUI Tasks | | | Combat Tasks | | |
|---|---|---|---|---|---|---|---|---|---|---|---|
| | | | Steps ↓ | Bo10 ↑ | ASR ↑ | Steps ↓ | Bo10 ↑ | ASR ↑ | Steps ↓ | Bo10 ↑ | ASR ↑ |
| ***Policies and VLA Models*** | | | | | | | | | | | |
| VPT (Baker et al., 2022) | - | End-to-end | 377 | 30.7 | $6.0^{\pm 11.4}$ | 398 | 6.7 | $0.8^{\pm 3.3}$ | 396 | 24.6 | $3.6^{\pm 7.7}$ |
| STEVE-1 (Lifshitz et al., 2024) | - | End-to-end | 384 | 29.4 | $8.0^{\pm 17.0}$ | 391 | 16.4 | $3.2^{\pm 8.4}$ | 395 | 14.7 | $3.9^{\pm 12.0}$ |
| ROCKET-1 (Cai et al., 2024a) | - | Workflow | 392 | 57.5 | $18.9^{\pm 24.3}$ | – | – | – | 320 | 63.9 | $27.9^{\pm 29.3}$ |
| JARVIS-VLA (Li et al., 2025a) | - | End-to-end | 305 | 55.3 | $30.0^{\pm 35.4}$ | 339 | 74.3 | $25.1^{\pm 23.9}$ | 352 | 61.9 | $18.5^{\pm 22.7}$ |
| ***VLM-based Hierarchical Agents*** | | | | | | | | | | | |
| LatentHA (Wang et al., 2024d) | Latent | Hierarchical | 363 | 54.2 | $24.4^{\pm 31.1}$ | 393 | 19.1 | $3.0^{\pm 7.5}$ | 371 | 24.6 | $8.5^{\pm 17.9}$ |
| MotionHA (Belkhale et al., 2024b) | Motion | Hierarchical | 336 | 51.0 | $27.4^{\pm 35.2}$ | – | – | – | 392 | 29.5 | $4.3^{\pm 10.8}$ |
| GroundingHA (Lee et al., 2025) | Grounding | Hierarchical | 290 | 55.0 | $37.1^{\pm 38.5}$ | 380 | 27.5 | $6.7^{\pm 10.8}$ | 346 | 76.3 | $26.5^{\pm 23.4}$ |
| SkillHA (Wang et al., 2024c) | Skill | Hierarchical | 365 | 31.3 | $11.3^{\pm 14.5}$ | 397 | 19.3 | $6.3^{\pm 9.2}$ | 393 | 12.8 | $6.5^{\pm 9.3}$ |
| ***Vision-Language-Action Models*** | | | | | | | | | | | |
| TextVLA | Text | End-to-end | 321 | 55.3 | $27.0^{\pm 17.0}$ | 291 | 75.4 | $25.8^{\pm 14.3}$ | 317 | 46.8 | $10.0^{\pm 6.1}$ |
| OpenHA | Text | End-to-end | 287 | 75.8 | $30.1^{\pm 13.9}$ | 314 | 82.6 | $32.5^{\pm 9.2}$ | 316 | 90.2 | $31.9^{\pm 13.7}$ |

stracted actions, formatted as $\{(o_t, A_t)\}_{t=1}^T$. (2) **Low-Level Action Data** ($D_a$)**:** The original trajectories containing only observations and primitive environmental actions, formatted as $\{(o_t, a_t)\}_{t=1}^T$. (3) **Chain-of-Action Data** ($D_{\mathbf{CoA}}$)**:** Trajectories containing the complete triplet of observation, generated high-level action, and the corresponding primitive action, formatted as $\{(o_t, A_t, a_t)\}_{t=1}^T$.

**Training Recipes.** All agents are initialized from Qwen2-VL-7B (Wang et al., 2024b) continual-trained on Minecraft-specific VQA and captioning data (Li et al., 2025a), with architecture-tailored training recipes for fair paradigm evaluation. Standard end-to-end VLA models are trained solely on the low-level action dataset $D_a$ to maximize $P(a_t \mid o_t, \text{ins})$; besides the Raw Action-based setting (following **JARVIS-VLA**), a TextVLA variant with Text Action-represented $a_t$ is additionally trained. For Hierarchical Agent (HA) baselines, the high-level VLM policy is trained on $D_A$ to learn $P(A_t \mid o_t, \text{ins})$, with low-level decoding policies trained separately per original implementations. Corresponding HAs are denoted as LatentHA, MotionHA, GroundingHA, and SkillHA based on the abstract action $A$ type. CoA-based agents adopt a two-stage curriculum: pretraining on $D_A \cup D_a$ (multi-task phase for familiarizing with abstract plan/primitive action vocabularies/formats), then fine-tuning exclusively on $D_{\text{CoA}}$ to $P(A_t, a_t \mid o_t, \text{ins})$ (Equation 2) for bridging planning and execution. $a_t$ is uniformly represented as Text Actions, yielding MotionCoA (Motion Actions as abstract actions) and GroundingCoA (Grounding Actions as abstract actions). The all-in-one generalist OpenHA agent uses curricula with $D_A$ and $D_{\text{CoA}}$ composed of a heterogeneous mixture of all action abstraction types considered. For fair comparison, VLA, all HA, and OpenHA models are trained with identical total computation.

**Training Details.** To ensure fairness, as mentioned throughout the paper, we keep the training settings as consistent as possible. All experiments were conducted on 32 NVIDIA A800-SXM4-80GB GPUs with Python 3.10 and CUDA 12.6. We adopt imitation learning using the AdamW optimizer with hyperparameters $\beta_1 = 0.9$, $\beta_2 = 0.95$, and $weight\_decay = 0.1$ to train Qwen2-VL-based models (Wang et al., 2024b). The batch size is fixed at 128. We employ a cosine learning rate schedule with a minimum learning rate of $1 \times 10^{-6}$, a maximum gradient norm of 1.0, and a fixed random seed of 42. This training setup first builds foundational knowledge and then enables the model to learn the crucial link between high-level actions and low-level execution. We use the TRL (von Werra et al., 2020) and VeOmni (Ma et al., 2025) libraries to train the agent models.

**Compared Models and Baselines.** Our primary analysis focuses on a controlled, systematic comparison of agents equipped with the different action spaces detailed in Section 3, including our proposed CoA-based hierarchical VLA models and OpenHA models, which are evaluated under the settings specified in Appendix C. To contextualize our results within the broader literature, we also compare against specialized policies that were trained on the VPT dataset. These baselines include the original VPT (Baker et al., 2022) and STEVE-1 (Lifshitz et al., 2024), etc.

### 4.2. Evaluation of Action Spaces in Hierarchical Agents

To identify the most effective action space for Hierarchical Agents (HAs), we conducted a controlled evaluation across our diverse, large-scale benchmark. Each VLM-based HA was trained on a dataset with an equivalent number of action tokens to ensure a fair comparison.

*Table 4.* Ablation study on the trade-off between inference mode and performance for agents using Motion and Grounding actions. We compare traditional Hierarchical Agents using a **Fast**, decoupled inference mode against our unified models trained with Chain of Action (CoA) and using the **Slow**, unified autoregressive mode. FPS denotes inference speed. The CoA models consistently achieve higher Success Rates at the cost of lower FPS.

| Model | Mode | FPS | Embodied | GUI | Combat |
|---|---|---|---|---|---|
| *Baseline* | | | | | |
| TextVLA | - | 1.36 | $27.0^{\pm 17.0}$ | $25.8^{\pm 14.3}$ | $10.0^{\pm 6.1}$ |
| *Motion Actions* | | | | | |
| MotionHA | Fast | 3.75 | $27.4^{\pm 35.2}$ | $0.0^{\pm 0.0}$ | $4.3^{\pm 10.8}$ |
| MotionCoA | Slow | 0.98 | $31.2^{\pm 14.1}$ | $28.7^{\pm 10.9}$ | $15.8^{\pm 4.2}$ |
| *Grounding Actions* | | | | | |
| GroundingHA | Fast | 5.61 | $37.1^{\pm 38.5}$ | $6.7^{\pm 10.8}$ | $26.5^{\pm 23.4}$ |
| GroundingCoA | Slow | 1.22 | $31.8^{\pm 12.5}$ | $30.1^{\pm 4.2}$ | $28.3^{\pm 9.9}$ |

*Table 5.* Performance comparison between specialist agents (trained on a single action space) and our universal OpenHA agent (trained on all action spaces). For each comparison, OpenHA is prompted to use the same inference format as the corresponding specialist (e.g., MotionCoA). The results show that the universally-trained OpenHA model outperforms the specialists, demonstrating positive knowledge transfer across action spaces.

| Model | Mode | Embodied | GUI | Combat |
|---|---|---|---|---|
| *Text Action* | | | | |
| TextVLA | Text | $27.0^{\pm 17.0}$ | $25.8^{\pm 14.3}$ | $10.0^{\pm 6.1}$ |
| OpenHA | Text | $\mathbf{27.5^{\pm 13.8}}$ | $\mathbf{32.5^{\pm 9.2}}$ | $\mathbf{31.9^{\pm 13.7}}$ |
| *Motion Actions* | | | | |
| MotionCoA | MotionCoA | $\mathbf{31.2^{\pm 14.1}}$ | $28.7^{\pm 10.9}$ | $15.8^{\pm 4.2}$ |
| OpenHA | MotionCoA | $24.8^{\pm 10.9}$ | $\mathbf{29.5^{\pm 8.7}}$ | $\mathbf{25.6^{\pm 9.8}}$ |
| *Grounding Actions* | | | | |
| GroundingCoA | GroundingCoA | $\mathbf{31.8^{\pm 12.5}}$ | $30.1^{\pm 4.2}$ | $28.3^{\pm 9.9}$ |
| OpenHA | GroundingCoA | $30.1^{\pm 13.9}$ | $\mathbf{35.1^{\pm 10.4}}$ | $\mathbf{29.8^{\pm 12.6}}$ |

Our first finding, detailed in Table 3, is that prior methods like STEVE-1 exhibit a significant performance drop on our benchmark (e.g., 8.0% ASR on Embodied tasks). This highlights their limited generalizability, as they were often trained on narrower task sets and struggle with diverse challenges like fishing or defeating creepers. In contrast, our VLM-based HAs demonstrate substantially stronger performance, underscoring the benefits of pre-trained models.

However, our central finding is that **no single action space is universally optimal; performance is highly contingent on the task domain**. This is illustrated in Figure 1 and supported by task-specific results: MotionHA achieves a nearly 100% success rate on "kill the sheep", while SkillHA is the top performer on "Chop down trees" with 95% success. The performance disparities stem from the inherent trade-offs in each action's expressive capabilities.

The agent using grounding actions, **GroundingHA**, is the most capable HA overall, achieving the highest ASR in both Embodied (37.1%) and Combat (26.5%) tasks. Its strength lies in explicitly binding an action (e.g., `mine`) to an entity's pixel coordinates, effectively leveraging the VLM's visual grounding ability. Its primary weakness is in partially observable scenarios where the target is off-screen. Conversely, the agent using object-agnostic motion primitives, **MotionHA**, excels at navigation-heavy Embodied tasks (27.4%) but completely fails in GUI tasks (0.0%). Its object-agnostic nature is a critical flaw, as it cannot specify *what* to interact with, rendering it incapable of precise, object-centric actions required for crafting. The other agents, **LatentHA** and **SkillHA**, show moderate but not leading performance in any single category.

### 4.3. Experiments on Chain-of-Action Training

Having validated task-specialized action abstractions, we test our core hypothesis: *can unifying these abstractions as intermediate reasoning steps in a single VLA boost overall performance?* We compare unified Chain-of-Action (CoA) models with traditional hierarchical agents (HA); our architecture supports Fast (Decoupled) mode (mimicking HA via separate lightweight decoder) and Slow (CoA) mode (autoregressive plan/action generation) for direct paradigm comparison (see Section 3.2).

Table 4 results show an inference efficiency-performance trade-off. For Motion Actions, MotionHA (3.75 FPS) is far more efficient than MotionCoA (0.98 FPS), yet MotionCoA triples Combat task ASR (4.3% → 15.8%), offsetting MotionHA's weakness in this domain. A similar trend holds for Grounding Actions, with Combat success rates rising from 26.5% to 28.3%. Our top CoA model, GroundingCoA, also outperforms the strong TextVLA baseline notably in Embodied (31.8% vs. 27.0% ASR) and Combat (28.3% vs. 10.0% ASR) domains. This confirms that generating abstract actions as intermediate "thoughts" structures the VLA's decision-making more effectively (see Appendix D for CoA operational details).

This reveals a novel inference-time scaling for autonomous agents: unlike standard training-time scaling (bigger models/more data), CoA provides a complementary improvement axis—allocating more inference-time compute to structured, explicit reasoning. By "thinking" more before acting, agents achieve higher performance, verifying that well-integrated abstract actions enhance VLA capabilities.

*Table 6.* Ablation experiments on the embodied environment Language Table (Lynch et al., 2023).

| Task | TextVLA | MotionHA | MotionCoA | OpenHA |
|---|---|---|---|---|
| BlockToBlock | $83^{\pm 9}$ | $76^{\pm 8}$ | $84^{\pm 9}$ | $87^{\pm 6}$ |
| BlockToRelLoc | $82^{\pm 5}$ | $83^{\pm 7}$ | $84^{\pm 7}$ | $87^{\pm 4}$ |
| SeparateBlocks | $96^{\pm 2}$ | $98^{\pm 1}$ | $97^{\pm 2}$ | $98^{\pm 1}$ |

### 4.4. Effectiveness of Training on Diverse Action Spaces

In this section, we try to answer: *Does training a single agent on a mixture of action spaces enable it to learn a more generalizable and effective decision-making policy?* We conduct a series of experiments comparing our universally-trained agent, OpenHA, against the specialist models to answer this questions. To ensure a fair comparison, we evaluate OpenHA by constraining its output to match the format of each specialist agent. See Appendix F for details. This allows us to isolate the effect of the training data diversity. The results of this comparison are presented in Table 5.

The data reveals that the universally-trained OpenHA agent outperforms the specialist models. For example, when using the MotionCoA inference format on Combat tasks, OpenHA achieves an ASR of 25.6% compared to the specialist MotionCoA's 15.8%. This consistent improvement across different inference formats and task categories strongly suggests that the model benefits from the diverse training data, learning a more robust and comprehensive internal policy. It shows that OpenHA has learned a generalized concept of object interaction from its exposure to Text and Grounding actions, and it can leverage this knowledge even when forced to express its intent through an object-agnostic action format. This confirms that the All-in-One training fosters a deeper understanding of tasks that transcends any single action representation. Suggest that co-training on diverse action spaces is a powerful mechanism for improving an agent's core reasoning and generalization capabilities.

### 4.5. Ablation on other environments

We also provide additional tests on the simpler embodied Language Table (Lynch et al., 2023) using Qwen2VL-7B as the base model. Original Language Table data (Lynch et al., 2023) was converted to Text Action `move(dx, dy)` and Motion Action `move left and push up`; other settings were consistent with OpenHA in Minecraft. We report step-200 success rates (%) per instruction (averaged over 200 rollouts), and Table 6 results are mean±stderr over 3 seeds.

Motion/Text Actions as abstract actions show task-dependent pros and cons: TextVLA (precise raw actions) outperforms in object-related tasks (e.g., block-to-block), while MotionHA excels in relative position tasks (e.g., block-to-relative location). Abstract actions

as thoughts boost agents' low-level action understanding (e.g., MotionCoA improved success in `BlockToBlock` and `BlockToRelative Location`). OpenHA, trained on multi-action spaces, outperformed all single-action agents across nearly all tasks, verifying the necessity and effectiveness of multi-class action joint training and generalization. Language Table experiments confirm VLM-based agents require task-specific action mastery.

## 5. Conclusions

In this work, we first conduct a large-scale analysis demonstrating that the optimal abstracted action for autonomous agents is highly task-dependent. To address this, we introduce the **Chain of Action** framework, which unifies high-level planning and low-level control within a single end-to-end VLA model. Building on this, we show that an **All-in-One** agent, trained on a diverse mixture of action spaces, learns a more generalizable policy and achieves superior performance compared to any specialist agent. Our findings suggest that synergizing multiple action representations, rather than selecting a single one, is a crucial step towards developing more capable and general-purpose autonomous agents.

## Impact Statement

This paper presents work whose goal is to advance the field of Machine Learning. There are many potential societal consequences of our work, none which we feel must be specifically highlighted here.

## Reproducibility Statement

Appendix B presents full details of the training setup (hardware, software versions and complete training recipe). Appendix B.2 explains the dataset construction and preprocessing, including our programmatic labeling pipeline. Table 9 summarizes the specific training recipes for each agent variant. A suite of specialist models trained under individual action spaces—namely MotionHA, GroundingHA, and TextVLA—alongside a unified **OpenHA** model jointly trained across multiple abstraction types, is planned for release. Complementing these models, we prepare large-scale datasets aligned with different action spaces: single-space datasets (Motion, Grounding, Text) and Chain-of-Action (CoA) datasets that integrate high-level reasoning with low-level execution. These resources are intended to enable fine-tuning and benchmarking, facilitate reproducibility, and accelerate progress in open-ended hierarchical agent research.

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

# A. Details of Abstracted Action Space used in DHA

We adopt multiple abstracted action spaces, including four core types: **Skill Actions**, **Grounding Actions**, **Motion Actions**, and **Latent Actions**. These abstract actions are represented as action tokens that guide the low-level policy during execution. Below, we provide details of each abstract action space and its semantics.

## A.1. Skill Action

Skill Actions (Driess et al., 2023) refer to semantically meaningful, atomic task segments that can serve as self-contained skill units within a larger instruction-following trajectory. In the context of long-horizon tasks, these actions act as high-level decompositions—each skill action represents a discrete subgoal that contributes to accomplishing the overarching instruction. For example, the instruction `Obtain a diamond` might be decomposed into a sequence of skill actions such as `Gather wood`, `Craft a pickaxe`, and `Mine stone`.

To construct a diverse and grounded set of skill actions, we begin by collecting discrete in-game events from the simulator environment (Guss et al., 2019). The events include common player interactions such as: *crafting an item*, *mining a block*, *killing an entity*, *dropping an item*, *using a tool*, and *interacting with blocks*.

These raw events are then provided to an LLM (Achiam et al., 2023), which autonomously clusters and merges semantically related events into higher-level skill task descriptions. This merging process is designed to reflect human-like reasoning about goal structures (e.g., "chop trees and collect logs" or "smelt iron ore into ingots").

Skill Actions are used during training and inference as shown in Table A.1, which illustrates a representative example.

---

**Example of Skill Action Data**

**Instruction**: Craft the magma block.
**Observation**:

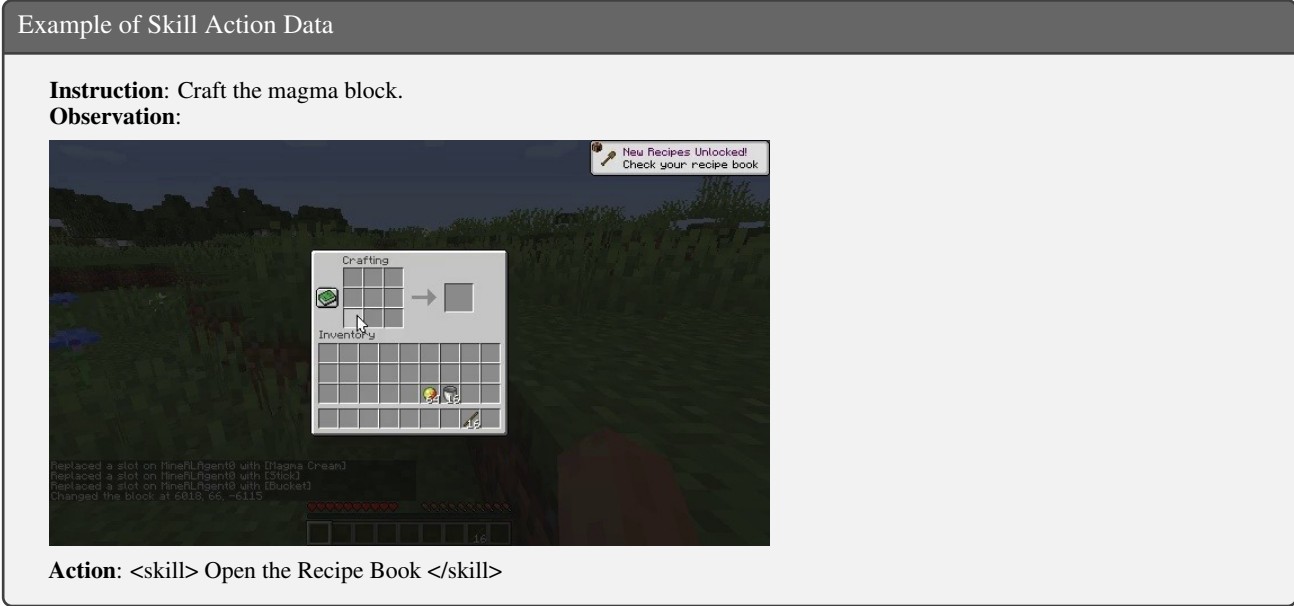

**Action**: <skill> Open the Recipe Book </skill>

---

## A.2. Grounding Actions

Grounding actions are designed to facilitate fine-grained interactions with visual targets in the environment by referencing specific spatial coordinates. When an object is visible and interactable in the current observation frame, we associate it with a point-based reference and assign a corresponding action type. If the object is not observable, the agent instead performs a general behavior (e.g., exploration).

To collect grounding action data, we follow a modified version of the procedure proposed in Jarvis-VLA (Li et al., 2025a). Specifically, we manually annotate the $(x, y)$ coordinates of interactable objects across 2,000 human gameplay trajectories. Each trajectory yields an average of 5 valid grounding samples. We further implement Backward Trajectory Relabeling (Cai et al., 2024a) to propagate the object references to earlier timesteps $(t - k)$, allowing for trajectory alignment. All relabeled coordinates are then manually verified, and erroneous or out-of-frame points are filtered out.

The full list of supported grounding actions and their formats is presented in Table 7.

*Table 7.* The grounding action space used in DHA.

| Index | Action | Format | Description |
|---|---|---|---|
| 1 | Kill | `Kill <\|point_start\|>(x,y)<\|point_end\|>` | Attack the mob located at coordinate $(x, y)$. The target must appear in the current frame. |
| 2 | Mine | `Mine <\|point_start\|>(x,y)<\|point_end\|>` | Destroy the block located at coordinate $(x, y)$. The target must appear in the current frame. |
| 3 | Approach | `Approach <\|point_start\|>(x,y)<\|point_end\|>` | Move toward the object at coordinate $(x, y)$. The object must be visible in the current observation. |
| 4 | Explore | `Explore` | Explore the environment when no specific target is visible in the current frame. |
| 5 | Right Click | `Right Click <\|point_start\|>(x,y)<\|point_end\|>` | Use the right-click interaction on the object at coordinate $(x, y)$. |
| 6 | Move To | `Move To <\|point_start\|>(x,y)<\|point_end\|>` | Move the cursor in the GUI to coordinate $(x, y)$ and interact. |
| 7 | No-op | `No-op` | Remain idle and perform no action. |

Below is an example of grounding action representation used during training and inference.

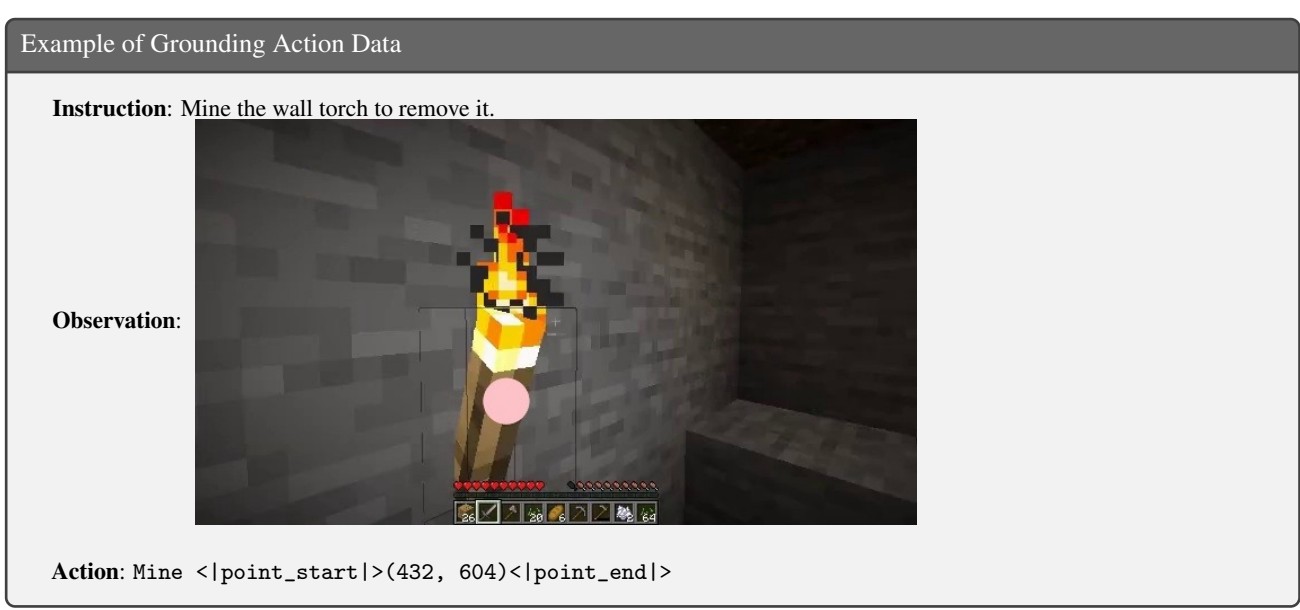

**Example of Grounding Action Data**

**Instruction**: Mine the wall torch to remove it.

**Observation**:

**Action**: Mine `<|point_start|>(432, 604)<|point_end|>`

### A.3. Motion Action

Motion actions are mid-level abstractions derived by analyzing the human action space defined in the Minecraft Wiki. To construct these, we follow a methodology similar to RT-H (Belkhale et al., 2024a), in which temporally and semantically adjacent raw actions are grouped into unified motion primitives. This abstraction enables more structured and interpretable behavior modeling while maintaining sufficient fidelity to the original control granularity.

By clustering similar low-level key or mouse inputs (e.g., repeated `W` presses and mouse movements for forward walking and camera turning), we generate compact representations such as `Turn left and down` or `Jump forward`. These motion-level actions serve as a bridge between high-level symbolic planning (e.g., grounding or skill actions) and low-level execution (raw keyboard/mouse events), and are directly executable by their corresponding motion policy within DHA's Mixture-of-Experts module.

The set of Motion Actions is shown in Table 8.

*Table 9.* Training recipes for different agent models.

| Model | HA | VLA | Hierarchical VLA | | OpenHA Model | |
| --- | --- | --- | --- | --- | --- | --- |
| Stage | 1 | 1 | 1 | 2 | 1 | 2 |
| Datasets | $D^A[GA|MA|SA]$ | $D^a$ | $D^A+D^a$ | $D^{\text{CoA}}$ | $D^A[MA|GA]+D^a$ | $D^{\text{CoA}}[MA+GA]$ |
| Consumed Tokens | 1.58B | 3.59B | 3.33B | 0.27B | 3.40B | 0.22B |
| Learning Rate | 5e-6 | 1e-5 | 1e-5 | 5e-6 | 1e-5 | 5e-6 |
| Training Steps | 4000 | 10400 | 10800 | 400 | 8200 | 300 |

*Table 8.* The basic Motion Action space we use in Minecraft.

| Index | Motion Action | Description |
| --- | --- | --- |
| 1 | <motion>turn left and down</motion> | Move the camera to the lower left. |
| 2 | <motion>turn right and down</motion> | Move the camera to the upper left. |
| 3 | <motion>turn left and up</motion> | Move the camera to the upper left. |
| 4 | <motion>turn right and up</motion> | Move the camera to the upper right. |
| 5 | <motion>attack / mine</motion> | Press and hold the left mouse button to attack or mine. |
| 6 | <motion>move forward</motion> | Press and hold the forward key to move forward. |
| 7 | <motion>jump</motion> | Jump. When swimming, it keeps the player afloat. |
| 8 | <motion>sprint</motion> | Move quickly in the direction of current motion. |
| 9 | <motion>use</motion> | Interact with the block. |
| 10 | <motion>hotbar.[1-9]</motion> | Selects the appropriate hotbar item from 1 to 9. |

The model can output combined motions by merging different motions mentioned above into the same <motion> tag and separating them with commas in sequence, for example: <motion>move forward, turn right and up</motion> means the combination of <motion>move forward</motion> and <motion>turn right and up</motion>.

Below is an example of motion action

---

**Example of Motion Action Data**

**Instruction**: Mine the white concrete block.
**Observation**:

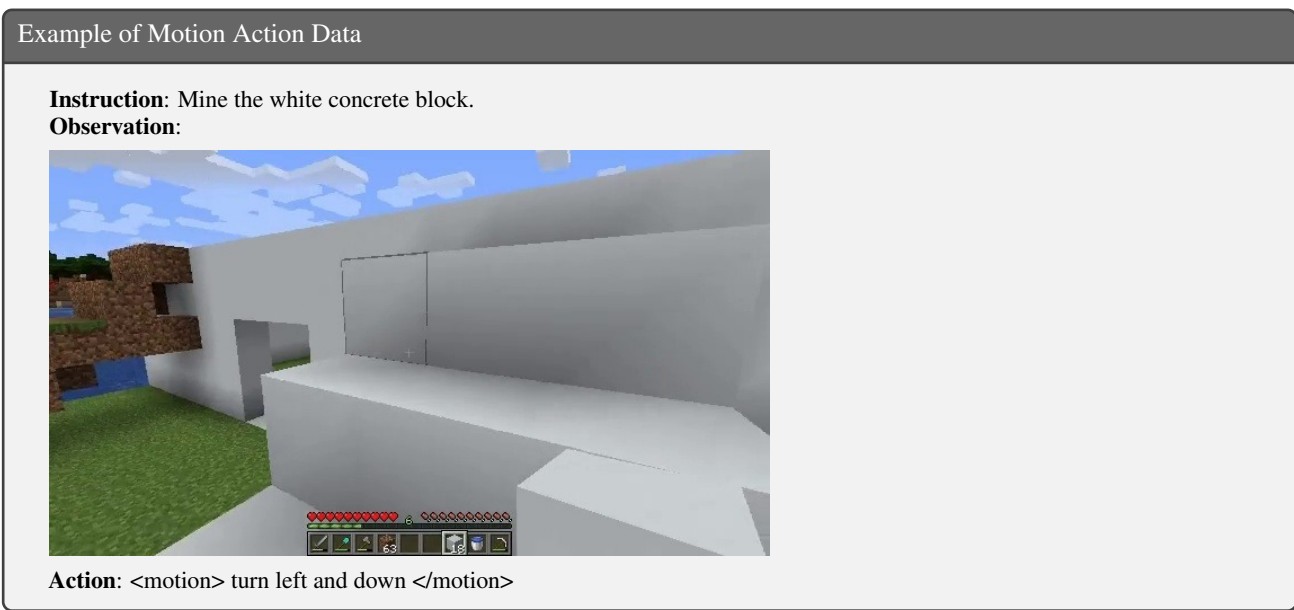

**Action**: <motion> turn left and down </motion>

---

# B. Details of Training

### B.1. Training Configuration

To ensure fairness, as mentioned throughout the paper, we keep the training settings as consistent as possible. All experiments were conducted on 32 NVIDIA A800-SXM4-80GB GPUs with Python 3.10 and CUDA 12.6. We adopt imitation learning using the AdamW optimizer with hyperparameters $\beta_1 = 0.9$, $\beta_2 = 0.95$, and $weight\_decay = 0.1$ to train Qwen2-VL-based models (Wang et al., 2024b). The batch size is fixed at 128. We employ a cosine learning rate schedule with a minimum learning rate of $1 \times 10^{-6}$, a maximum gradient norm of 1.0, and a fixed random seed of 42. This training setup first builds foundational knowledge and then enables the model to learn the crucial link between high-level actions and low-level execution. We use the TRL (von Werra et al., 2020) and VeOmni (Ma et al., 2025) libraries to train the agent models.

### B.2. Construction of Datasets

Our raw training data is derived from the OpenAI Video Pre-Training (VPT) dataset (Baker et al., 2022), which contains a large collection of expert trajectories in Minecraft. However, the original VPT dataset only provides pairs of visual observations and corresponding low-level environmental actions $(o_t, a_t)$. To enable the training of agents with high-level action spaces, we developed a programmatic labeling pipeline. This pipeline applies a set of rule-based heuristics to transform the raw low-level action sequences into their corresponding high-level abstracted representations $(A_t)$, including Latent, Motion, Grounding, and Language Skill actions.

The detailed training parameters and datasets are summarized in Table 9.

# C. Details of Inference

We standardize the output format of all agentic models. For each interaction, both the CoA model and the hierarchical agentic model are restricted to producing a single executable instruction. The complete output format is shown below.

```
<|im_start|>system\nYou are a helpful assistant.<|im_end|>
<|im_start|>user\n[system_prompt][instruction][observation_1]<|im_end|>
<|im_start|>assistant\ n[Action_1]<|im_end|>
...
<|im_start|>user\n[observation_t]<|im_end|>
<|im_start|>assistant\n
```

For CoA models, this constraint means that only one environment action can be produced at each step. For hierarchical models, whenever the high-level VLM outputs an abstract action, the corresponding action policy is allowed to execute up to four instructions.

During inference, we allow agentic models to maintain their own working memory. Specifically, the VLM can retain a trajectory of up to 15 steps, keeping track of its previous outputs together with the observed environmental information.

We adapt vLLM (Kwon et al., 2023) to accelerate inference speed.

# D. Case Study on CoA Models

In this section, we present a detailed case study to evaluate the performance of the OpenHA model using the Chain-of-Action (CoA) framework. We selected four successful rollouts performed by OpenHA, two from the **GroundingCoA Inference Format** and two from the **MotionCoA Inference Format**.

As discussed in Section 3 , each task begins with the model generating an abstract action, which serves as an intermediate reasoning step, followed by the model outputting the corresponding **primitive action**. To improve interpretability and reduce the complexity of displaying hundreds of interactions, we use a time-adjacent clustering strategy. This method groups consecutive, similar abstract actions that occur within a specific time window. For example, in the **GroundingCoA mode**, if the model outputs the same abstract action (e.g., "move camera") repeatedly over several time steps, we consider these actions part of the same grounding action as long as the Euclidean distance between action points remains below a threshold

(e.g., 50 pixels). In the **MotionCoA mode**, abstract actions are grouped together if the action type remains consistent over time.

Rather than displaying each individual step, we group the primitive actions generated for a particular abstract action during a time slice. For instance, if the abstract action is "turn right," and the model repeatedly generates similar low-level actions (such as moving the camera), we aggregate these primitive actions into a single **action sequence**. These grouped sequences are then analyzed to assess task success.

Figures 2, 3, 4, and 5 illustrate the model's execution for each task, with both the abstract and primitive actions performed at each step.

From these case studies, we observe that an abstract action often corresponds to a specific pattern of primitive actions. For example, when the grounding action is 'move camera to (x,y)', the primitive action sequence typically begins with several mouse movements followed by a series of mouse clicks. This observation suggests that the CoA framework operates as a two-phase reasoning process: first, the model decides on the **abstract action**, and then it generates the **primitive actions** based on the abstract plan. In other words, CoA models decompose tasks into abstract actions, which are then broken down into executable actions, facilitating a clear separation between reasoning and action execution.

This two-stage process bears resemblance to the Chain-of-Thought framework, where intermediate reasoning steps guide the model's eventual execution of low-level tasks. In this analogy, the abstract action can be thought of as a "rationale," similar to the "thought" process in CoT. However, instead of purely reasoning through abstract steps in isolation, CoA uses this intermediate action to directly inform the subsequent low-level actions, effectively bridging high-level planning with executable action in an integrated manner.

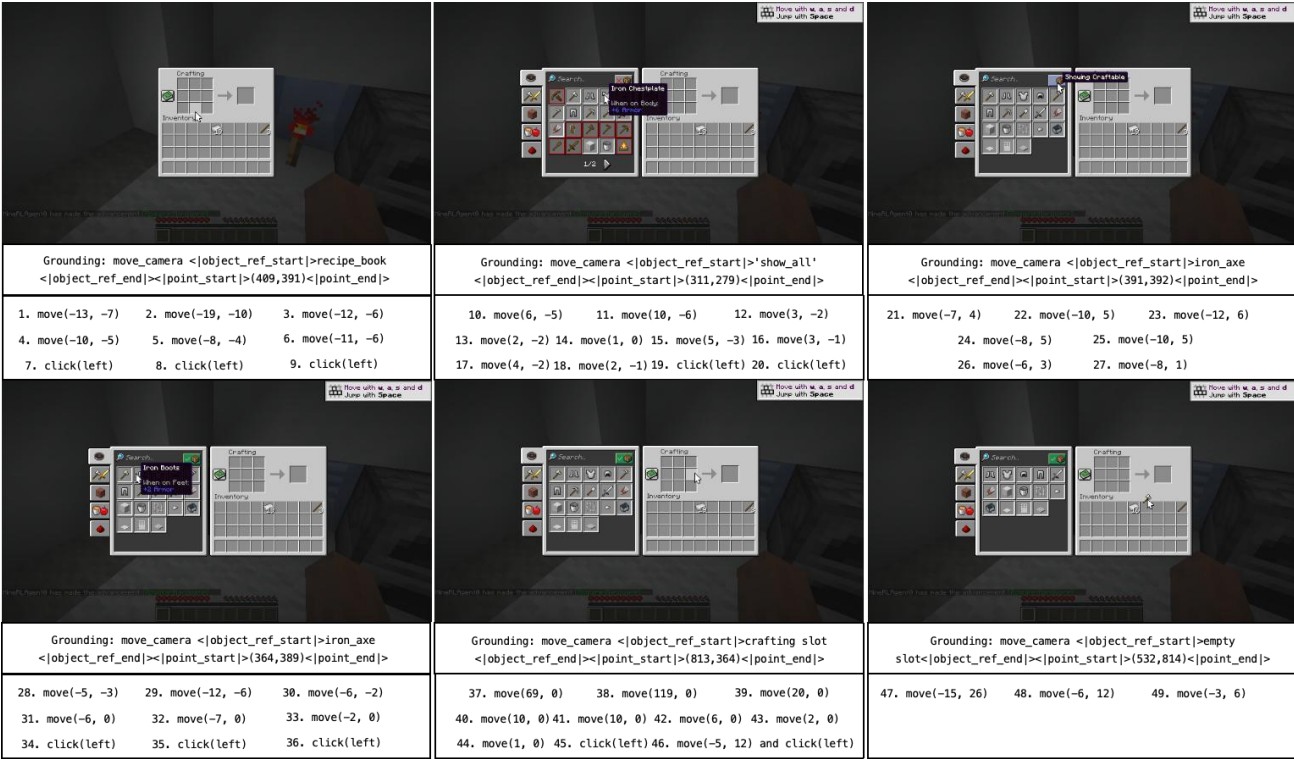

*Figure 2.* Execution of the "Craft an Iron Sword" task using the GroundingCoA inference format.

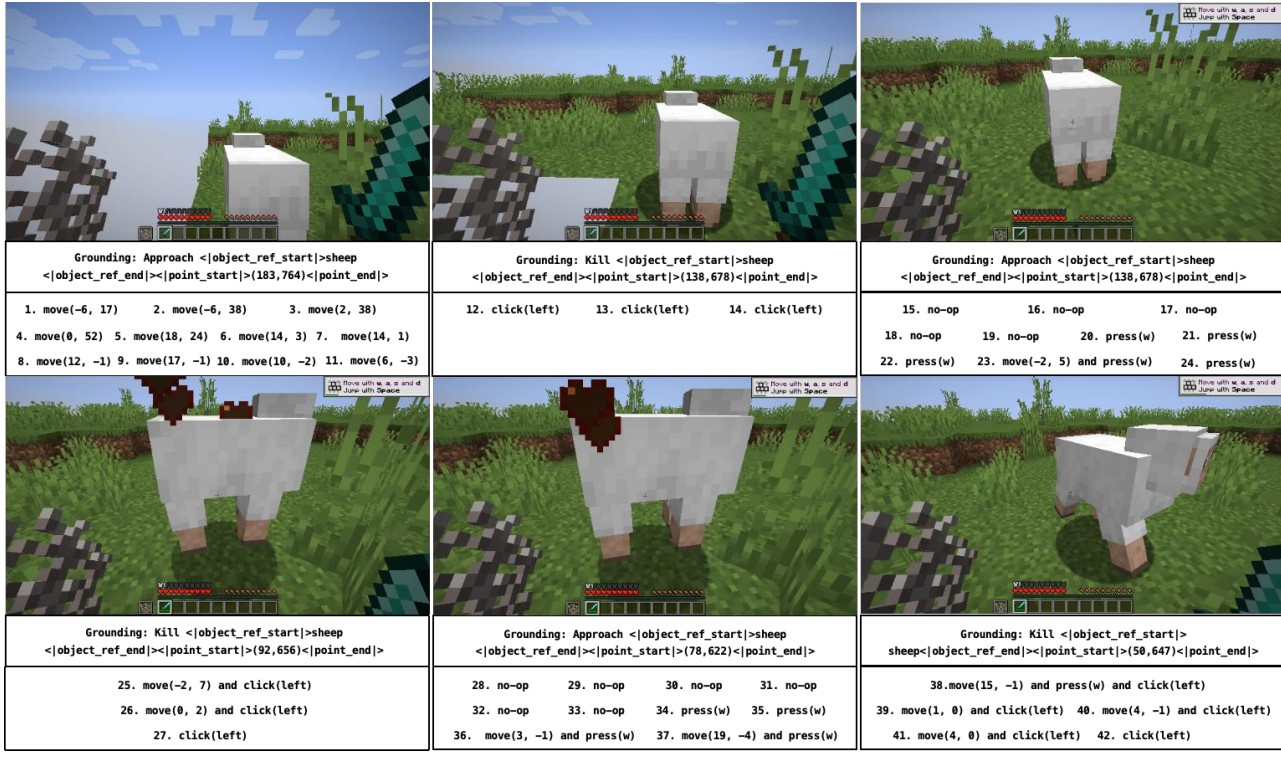

*Figure 3.* Execution of the "Kill the Sheep" task using the GroundingCoA inference format.

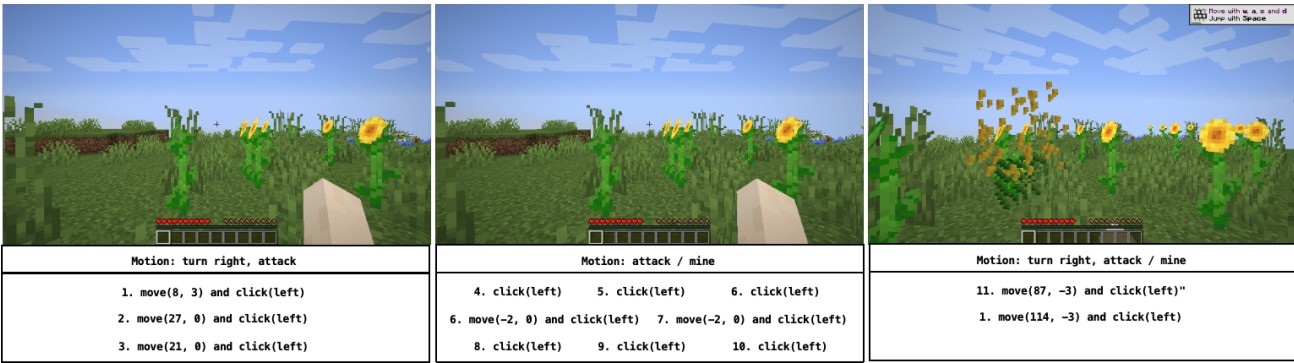

*Figure 4.* Execution of the "Mine a Sunflower" task using the MotionCoA inference format.

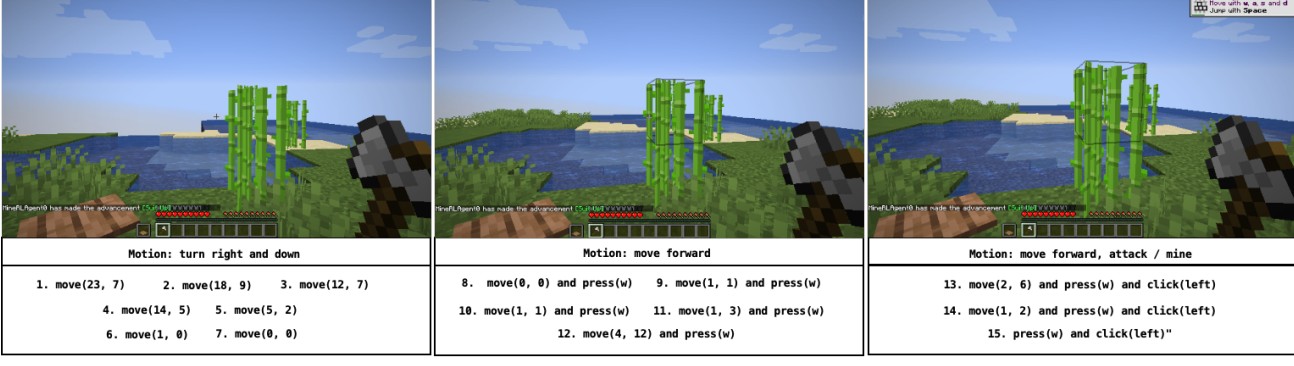

*Figure 5.* Execution of the "Mine a Sugar Cane" task using the MotionCoA inference format.

# E. Additional Experiments and Analysis

In this section, we deepen our empirical analysis of the Chain-of-Action (CoA) framework through a comprehensive set of ablation studies and generalization tests. We specifically investigate the impact of primitive action representations and provide a failure analysis of high-level skill abstractions. Furthermore, we verify the robustness of our prompting formats, demonstrate the advantages of CoA over Chain-of-Thought reasoning, analyze inference efficiency trade-offs, and finally confirm the generalizability of our approach to MoE architectures.

## E.1. Ablation on Primitive Action Representations

We assess the efficacy of our default Text Action (TA) encoding by comparing it against the Raw Action (RA) encoding strategy, which is widely adopted in prior works such as OpenVLA.

**Experimental Setup.** We implemented two distinct representations for the primitive environmental action space ($\mathcal{A}_{env}$) under identical training constraints:

- **Raw Action (RA):** Discretizes continuous controls (e.g., mouse movement) into bins mapped to reserved tokens (e.g., `<action_1>`). These tokens are initialized randomly outside the original vocabulary and lack inherent semantic meaning.

- **Text Action (TA):** Serializes actions into textual function calls (e.g., `move(30, 0)`). This approach reuses the foundation model's existing subword vocabulary.

**Results and Analysis.** As illustrated in Table 10, TextVLA consistently outperforms RawVLA, yielding an improvement of approximately 2.3% in Average Success Rate (ASR) across all categories. We attribute this performance gap to *semantic alignment* and *data efficiency*.

RA tokens require the model to learn physical relationships (such as the ordinal magnitude of movements) entirely from scratch. In contrast, TA leverages the VLM's pre-trained world knowledge—for instance, the model inherently understands that the string "30" represents a larger magnitude than "10". This **semantic prior** allows the agent to focus its capacity on policy learning rather than basic symbol grounding, leading to superior generalization, particularly in regimes with limited demonstration data.

*Table 10.* **Comparison of Primitive Action Representations.** We report the Average Success Rate (ASR) across three task categories. TextVLA consistently outperforms RawVLA, demonstrating the benefit of leveraging pre-trained semantic knowledge for low-level control.

| Model | Embodied ASR | GUI ASR | Combat ASR |
|---|---|---|---|
| RawVLA | 24.1 | 22.3 | 9.4 |
| TextVLA | **27.0** | **25.8** | **10.0** |

## E.2. Analysis of Skill-Level Abstraction

While Motion and Grounding actions serve as effective intermediate representations in our framework, high-level **Skill Actions** (e.g., "gather wood", "craft a pickaxe") are theoretically appealing due to their resemblance to human-like task decomposition, as seen in works like SayCan. To rigorously investigate their efficacy in our domain, we implemented a Skill-based Hierarchical Agent (SkillHA) and evaluated it on the full OpenHA benchmark.

As shown in Table 11, SkillHA achieves success rates of only 11.3%, 6.3%, and 6.5% across the three task categories. It consistently underperforms both the flat TextVLA baseline and other hierarchical variants (MotionHA and GroundingHA), indicating a fundamental limitation in this specific abstraction for our tasks.

We conducted a qualitative case study to investigate the root cause of this failure. We observed that Skill Actions are often too abstract relative to the immediate control granularity required for the OpenHA benchmark tasks.

For instance, in a combat task initialized with "kill a zombie", the high-level VLM planner frequently outputs a skill action that merely paraphrases the goal (e.g., `Skill: kill the zombie`). In such scenarios, the abstract action fails to provide

*Table 11.* **Performance of Skill-Level Abstraction.** Comparing SkillHA against other hierarchical and flat baselines. SkillHA significantly underperforms, suggesting that skill-level descriptions are too abstract to bridge the gap between instruction and execution.

| Model | Embodied ASR | GUI ASR | Combat ASR |
|---|---|---|---|
| TextVLA (Flat) | 27.0 | **25.8** | 10.0 |
| MotionHA | 27.4 | 0.0 | 4.3 |
| GroundingHA | **37.1** | 6.7 | **26.5** |
| SkillHA | 11.3 | 6.3 | 6.5 |

any information gain. It offers neither spatial guidance (unlike Grounding, which provides coordinates) nor dynamic control cues (unlike Motion). Consequently, the low-level policy receives no useful intermediate signal to bridge the semantic gap between the high-level instruction and the primitive execution.

**Conclusion.** This negative result reinforces our core finding: the optimal abstraction level is highly task-dependent. While skill actions may be beneficial for long-horizon compositional planning (e.g., "build a house"), they lack the precision required for the tactical control tasks evaluated in this study. This validates our design choice to prioritize Motion and Grounding actions for the CoA framework.

### E.3. Robustness to Abstract Action Formats

A potential concern regarding the Chain-of-Action framework is whether its performance relies on fitting a specific textual template or prompt structure. To investigate this, we conducted an ablation study comparing our standard natural language format against a structured XML-based format for representing abstract actions.

**Experimental Setup.** We trained two variants of the GroundingCoA agent using identical data and hyperparameters, differing only in the string formatting of the intermediate abstract action:

- **Text Format (Default):** Uses a natural language style representation, e.g., `Grounding: approach(point=[201, 105])`.

- **XML Format:** Uses a structured tag-based representation, e.g., `<grounding><func>approach</func>...`.

*Table 12.* **Sensitivity to Abstract Action Formats.** Comparison of agent performance using GroundingCoA with different output schemas. The marginal difference confirms that CoA is robust to format variations.

| Format | Example Schema | Embodied ASR | Combat ASR |
|---|---|---|---|
| Text | `Grounding: approach(point=[...])` | $31.3 \pm 1.6$ | $29.1 \pm 1.0$ |
| XML | `<grounding><func>approach</func>...` | $32.1 \pm 1.2$ | $28.9 \pm 0.7$ |

**Results and Analysis.** As shown in Table 12, the performance difference between the two formats is negligible. The XML format achieves a slightly higher success rate on Embodied tasks (32.1% vs. 31.3%) but a comparable rate on Combat tasks (28.9% vs. 29.1%).

This finding suggests that the efficacy of the CoA framework stems from the high-level *understanding of action semantics* rather than memorization of syntactic patterns. The abstract action serves as a robust semantic anchor that guides low-level execution, regardless of whether it is wrapped in natural language or structured tags. This robustness to formatting simplifies the prompt engineering process, allowing future work to adopt whichever format is most convenient for parsing or integration.

### E.4. Comparison with Chain-of-Thought Reasoning

To demonstrate that the benefit of the Chain-of-Action framework extends beyond standard reasoning techniques, we compared it against a Chain-of-Thought (CoT) baseline.

**Experimental Setup.** We trained a CoT baseline using the same backbone and data scale as our CoA models. In the CoT setup, the agent generates a free-form natural language rationale (e.g., "I need to approach the sheep to attack it") before predicting the primitive action, rather than generating a structured abstract action.

*Table 13.* **CoA vs. Chain-of-Thought (CoT).** CoA significantly outperforms the natural language reasoning baseline (CoT). This indicates that structured abstract actions provide more precise, actionable guidance for low-level control than free-form language thoughts.

| Reasoning Method | Embodied ASR | Combat ASR |
|---|---|---|
| VLA | 23.9 | 27.1 |
| Chain-of-Thought (CoT) | 27.2 | 22.4 |
| CoA (OpenHA-MotionCoA) | 29.7 | 26.3 |
| CoA (OpenHA-GroundingCoA) | **35.5** | **30.6** |

**Results and Analysis.** As shown in Table 13, both MotionCoA and GroundingCoA outperform the CoT baseline. Notably, CoT performs worse than the baseline in Combat tasks (22.4% vs 27.1%). We attribute this to the *abstraction gap*: while natural language is effective for high-level planning, it lacks the precision required for immediate low-level control (e.g., specific mouse dynamics). In contrast, CoA's abstract actions serve as a functional intermediate representation that is tightly aligned with the execution space, effectively bridging the gap between intent and action.

### E.5. Inference Efficiency and Dynamic Switching

A key consideration for hierarchical models is the trade-off between reasoning depth (performance) and computational cost (inference speed). We investigated the inference FPS and performance across different operating modes.

Beyond the fixed "Fast" (Decoupled) and "Slow" (Unified Autoregressive) modes, we evaluated an **Auto** mode where the agent autonomously decides per step whether to follow the full CoA generation path or output a primitive action directly. This behavior emerges naturally from the autoregressive policy without explicit supervision.

*Table 14.* **Inference Efficiency Trade-offs.** Comparison of Fast, Slow, and emergent Auto modes using MotionCoA. The Auto mode achieves a balance, maintaining high success rates comparable to Slow mode while operating significantly faster.

| Inference Pattern | Embodied ASR | Speed (FPS) |
|---|---|---|
| Fast Mode (Decoupled) | $27.4 \pm 3.5$ | **5.61** |
| Slow Mode (Unified) | $\mathbf{31.2 \pm 1.4}$ | 1.22 |
| Auto Mode (Emergent) | $30.7 \pm 1.4$ | 4.96 |

**Results and Analysis.** Table 14 reveals a significant trade-off: Slow mode maximizes performance (31.2% ASR) but at a cost of lower throughput (1.22 FPS). However, the emergent **Auto mode** achieves a "best of both worlds" result, reaching near-optimal performance (30.7% ASR) with a 4x speedup (4.96 FPS) compared to Slow mode. This demonstrates that the CoA agent allows for dynamic computation allocation at runtime based on task complexity.

### E.6. Generalization to MoE Architectures

To ensure that our findings are not specific to the dense architecture of Qwen2-VL, we evaluated the CoA framework on a Mixture-of-Experts (MoE) backbone, specifically **Qwen3-VL-30B-A3B**.

*Table 15.* **Generalization to MoE Architecture.** CoA maintains its performance advantage over standard VLA baselines when scaled to the Qwen3-VL (MoE) architecture, confirming that the framework is architecture-agnostic.

| Format | Base LM | Embodied ASR | Combat ASR |
|---|---|---|---|
| TextVLA | Dense (Qwen2-7B) | $23.9 \pm 8$ | $27.1 \pm 11$ |
| GroundingCoA | Dense (Qwen2-7B) | $31.3 \pm 16$ | $29.1 \pm 10$ |
| TextVLA | MoE (Qwen3-30B) | $32.1 \pm 19$ | $34.9 \pm 16$ |
| GroundingCoA | MoE (Qwen3-30B) | $\mathbf{37.8 \pm 12}$ | $\mathbf{37.8 \pm 10}$ |

**Results and Analysis.** As shown in Table 15, the GroundingCoA agent consistently outperforms the TextVLA baseline on the MoE architecture, achieving gains of +5.7% and +2.9% in Embodied and Combat tasks, respectively. This confirms that the effectiveness of the Chain-of-Action framework generalizes across different model architectures (Dense vs. MoE) and scales effectively with model capacity.

## F. OpenHA Output Format Constraints

Foundation models possess strong instruction-following abilities (Brown et al., 2020). We leverage different contexts to control the output format of OpenHA. For example, if we want OpenHA to produce outputs in the same format as MotionCoA, we must specify in the system prompt that OpenHA should generate actions in a "first motion, then text action" structure. For decision-making models, a well-designed system prompt should include the agent's role-play identity, the assigned task, the action space it must adhere to, and the organizational form of the chain of action. In this section, we introduce the system prompts used by OpenHA across different modalities.

---

**System Prompt for TextVLA-like Output**

```
You are an AI agent performing tasks in Minecraft based on given instructions, action history,
    and visual observations (screenshots). Your goal is to take the optimal action to complete
    the task.

## Output Format
Action: ...

## Action Space
* move('dx', 'dy') # Move the mouse position; dx and dy represent horizontal and vertical
    movement, respectively.
* click('left' or 'right')) # left click or right click the mouse
    - left_click # Attack; In GUI, pick up the stack of items or place the stack of items in a
        GUI cell; when used as a double click (attack - no attack - attack sequence), collect
        all items of the same kind present in inventory as a single stack.
    - right_click # Place the item currently held or use the block the player is looking at. In
        GUI, pick up the stack of items or place a single item from a stack held by mouse.
* press(keys) # press the keyboard buttons
    - 'w' # Move forward.
    - 's' # Move backward.
    - 'a' # Strafe left.
    - 'd' # Strafe right.
    - 'e' # Open or close inventory and the 2x2 crafting grid.
    - 'space' # Jump.
    - 'q' # Drop a single item from the stack of items the player is currently holding. If the
        player presses ctrl+q then it drops the entire stack. In the GUI, the same thing
        happens except to the item the mouse is hovering over.
    - '1'-'9' # Switch active item to the one in a given hotbar cell.
    - 'left.control' # Move fast in the current direction of motion.
    - 'left.shift' # Move carefully in current direction of motion. In the GUI it acts as a
        modifier key: when used with attack it moves item from/to the inventory to/from the
        hotbar, and when used with craft it crafts the maximum number of items possible
        instead of just 1.
* no_op # wait and do not interact with the world

If multiple actions are activated, use ', ' connect.

## User Instruction
{instruction}
```

---

**System Prompt for MotionHA-like Output**

```
You are an AI agent performing tasks in Minecraft based on given instructions and visual
    observations (screenshots). Your goal is to take the next optimal action to complete the
    task.

## Action Space
* move <direction> # Move in one or more of the following directions: forward, backward,
    right, left. You may combine directions (e.g., forward and left).
* sprint # Sprint in the current movement direction.
* sneak # Sneak; modifies movement in GUI and world.
* turn <direction> # Turn the camera view. Supported directions: up, down, right, left, or
    combinations like left and down.
* cursor move <direction> # Move the GUI cursor. Supported directions: up, down, right, left,
    or combinations like right and up.
* jump
* drop # Drop one item; Use Ctrl + drop to release the full stack.
* swap # Swap items between hands.
* switch hotbar # Switch to a different hotbar slot.
* open inventory
* close inventory
* close gui
* attack / mine # Use the main hand to attack or break a block.
* place / use # Use or place the held item.
* operate gui # Click or operate an element in the GUI.
* choose

If multiple actions are activated, use ', ' connect.

## Continuously take action until the task is completed.
Motion: motion action
## User Instruction
{instruction}
```

System Prompt for GroundingHA-like Output

You are an AI agent performing tasks in Minecraft based on given instructions, action history,
    and visual observations (screenshots). Your goal is to take the next optimal action to
    complete the task.

# Action
## You have the following actions.
* Kill <|point_start|>(x,y)<|point_end|> # Kill the Minecraft mobs marked by the coordinate
    (x, y). Note: The target of Action Kill needs to appear in the current observation image.
* Mine <|point_start|>(x,y)<|point_end|> # Destroy the Minecraft blocks marked by the
    coordinate (x, y). Note: The target of Action Mine needs to appear in the current
    observation image.
* Approach <|point_start|>(x,y)<|point_end|> # Approach to the target object marked by the
    coordinate (x, y). Note: The target of Action Approach needs to appear in the current
    observation image.
* Explore # Explore the environment. Note: The target of Action Explore should not appear in
    current observation image.
* right click <|point_start|>(x,y)<|point_end|> # Use the right click button to interact with
    the object marked by the coordinate (x, y).
* move_camera <|point_start|>(x,y)<|point_end|> # Move the cursor in the GUI to the location
    of the object identified by the coordinate (x, y).
* no-op # Wait and do not interact with the world.

If multiple actions are activated, use ', ' connect.

## Continuously take action until the task is completed.
Grounding: grounding actions
## User Instruction
{instruction}

System Prompt for MotionCoA-like Output

You are an AI agent performing tasks in Minecraft based on given instructions, history, and
    visual observations (screenshots). Your goal is to take the next optimal action to
    complete the task.
## Action Space
Your action space is hierarchical, including motion-level and raw-action. You need to output a
    hierarchical action chain until the final raw-action is output. The action spaces at each
    level are as follows:

### motion action

* move <direction>
* sprint / sneak
* turn <direction>
* cursor move <direction>
* jump
* drop
* swap
* switch hotbar
* open / close inventory
* inventory
* close gui
* attack / mine
* place / use
### raw actions

* move('dx', 'dy') # Move the mouse position; dx and dy represent horizontal and vertical
    movement, respectively.
* click('left' or 'right') # left click or right click the mouse
    - left_click # Attack;
    - right_click # Place the item currently held or use the block the player is looking at.
* press(keys) # press the keyboard buttons
    - 'w' # forward W key Move forward.
    - 's' # Move backward.
    - 'a' # Strafe left.
    - 'd' # Strafe right.
    - 'e' # Open or close inventory and the 2x2 crafting grid.
    - 'space' # Jump.
    - 'q' # Drop a single item from the stack of items the player is currently holding.
    - '1'-'9' # Switch active item to the one in a given hotbar cell.
    - 'left.control' # Move fast in the current direction of motion.
    - 'left.shift' # Move carefully in current direction of motion.
* no_op # wait and do not interact with the world
If multiple actions are activated, use and connect.
## Continuously take action until the task is completed.
Wrap your hierarchical action:
Motion: motion actions
Action: raw actions
## User Instruction
{instruction}

System Prompt for GroundingCoA-like Output

```
You are an AI agent performing tasks in Minecraft based on given instructions, history, and
    visual observations (screenshots). Your goal is to take the next optimal action to
    complete the task.
## Action Space
Your action space is hierarchical, including grounding-level and raw-action. You need to
    output a hierarchical action chain until the final raw-action is output. The action spaces
    at each level are as follows:

### grounding actions

* Kill <|object_ref_start|>mob<|object_ref_end|>
    <|point_start|>(x,y)<|point_end|> # Kill the Minecraft mobs.
* Mine <|object_ref_start|>block<|object_ref_end|>
    <|point_start|>(x,y)<|point_end|> # Destroy the Minecraft blocks.
* Approach <|object_ref_start|>object<|object_ref_end|>
    <|point_start|>(x,y)<|point_end|> # approach to the target object.
* right click <|point_start|>(x,y)<|point_end|> # use the right lick button to interact with
    the object.
* move to <|point_start|>(x,y)<|point_end|> # Move the cursor in the GUI to the location of
    the object.
* no-op

### raw actions
* move('dx', 'dy') # Move the mouse position; dx and dy represent horizontal and vertical
    movement, respectively.
* click('left' or 'right') # left click or right click the mouse
    - left_click # Attack;
    - right_click # Place the item currently held or use the block the player is looking at.
* press(keys) # press the keyboard buttons
    - 'w' # forward W key Move forward.
    - 's' # Move backward.
    - 'a' # Strafe left.
    - 'd' # Strafe right.
    - 'e' # Open or close inventory and the 2x2 crafting grid.
    - 'space' # Jump.
    - 'q' # Drop a single item from the stack of items the player is currently holding.
    - '1'-'9' # Switch active item to the one in a given hotbar cell.
    - 'left.control' # Move fast in the current direction of motion.
    - 'left.shift' # Move carefully in current direction of motion.
* no_op # wait and do not interact with the world
If multiple actions are activated, use and connect.

## Continuously take action until the task is completed.
Wrap your hierarchical action:
Grounding: grounding actions
Action: raw actions
## User Instruction
{instruction}
```

*Table 16.* Task configurations in the combat, embodied, and GUI task categories.

| Task name | Seed | Coordinates | Resources provided | Language instruction |
|---|---|---|---|---|
| kill_entity:sheep | 3001859308298079279 | [31, 64, 61] | Diamond sword | Kill the sheep. |
| kill_entity:slime | 3001859308298079279 | [31, 64, 61] | Diamond axe | Combat the slime. |
| mine_block:bamboo | 1 | [170, 65, 69] | None | Mine bamboo with your hand. |
| mine_block:glass | 1997 | [2554, 74, 1837] | Iron axe | Mine the glass. |
| craft_item:diamond_hoe | 1 | [-1838, 72, 30601] | 2 sticks, 2 diamonds | Craft a diamond hoe. |
| craft_item:glass_bottle | 1 | [-1838, 72, 30601] | 3 glass | Craft a glass bottle with glass. |

## G. OpenHA Benchmark

The open-ended game *Minecraft* provides an excellent foundation for studying different types of action spaces. Building upon the environment configuration of Lin et al. (2023), we design a benchmark of 800 diverse task settings, organized into three major categories: *embodied tasks*, *GUI tasks*, and *combat tasks*. Each agentic model is evaluated on every task with three independent trials, and we record both the number of steps required to complete the task and the success rate. Table 16 presents an overview of representative task configurations.

