# OpenReview forum: "OpenHA: A Series of Open-Source Hierarchical Agentic Models in Minecraft"
_ICML.cc/2026/Conference — ICML 2026 regular_

### Official Review · Reviewer_ka9T · 2026-02-25

**Soundness:** 3
**Presentation:** 3
**Significance:** 2
**Originality:** 2
**Overall Recommendation:** 3
**Confidence:** 4

**Summary:**

This paper introduces OpenHA, studying how the choice of action space affects the performance of Minecraft agents. Technically, the paper introduces Chain-of-Action (CoA) and the OpenHA suite, aiming to address the challenge of defining optimal action spaces in AI agents. The authors demonstrate that an "All-in-One" agent, trained on a mixture of action spaces, can outperform specialized baselines in Minecraft.

**Compliance With Llm Reviewing Policy:**

Affirmed.

**Key Questions For Authors:**

1. **Is A genuinely “reasoning” or just an auxiliary label?** Can you provide evidence that the intermediate abstract action contributes causally, e.g., intervene on A (swap A between states, perturb coordinates, replace with plausible alternatives) and measure predictable changes in low-level behavior? Without such interventions, “reasoning” claims feel under-verified.
2. **When to use fast vs slow mode ?** Can you provide a principled criterion (or learned policy) for switching modes (controllability under which condition), or at least task-feature analysis predicting when CoA’s slow mode is worth the FPS cost? This would strengthen the “inference-time scaling” claim.

**Limitations:**

No, at least not in the main text. The limitations are not adequately discussed given several important risks: (1) potential programmatic labeling leakage and reliance on heuristic relabeling pipelines, (2) the “reasoning” interpretation of abstract actions is not rigorously validated (needs intervention/faithfulness tests), (3) results are primarily shown on a single main backbone and a single main environment family, and (4) large benchmark/task releases could lead to overfitting or benchmark gaming unless strong OOD splits and evaluation protocols are emphasized. The paper should explicitly acknowledge these points and clarify reproducibility details for the labeling pipeline and training recipes.

**Strengths And Weaknesses:**

## Strengths
1. The paper tackles a real, practical bottleneck: action interface design for embodied/VLA agents. The claim that “no single action space is universally best” is plausible, and the broad task coverage helps support it.
2. CoA is simple and implementable, and the fast vs. slow inference experiments cleanly expose an important trade-off (speed vs. success).
3. The 800-task, hand-designed/verified benchmark is potentially valuable, especially given instability concerns with LLM-generated tasks.

## Weaknesses
1. **CoA looks more like a token or prompt decomposition** than a substantive algorithmic contribution. Eq. (2) is essentially the chain rule over a concatenated sequence (“generate A, then generate a|A”), closely resembling standard “plan-then-act or CoT” formatting. There is probably no new optimization insight, and no explicit mechanism that enforces A to be useful or faithful. As a result, the method risks being viewed as a supervision or formatting choice rather than a new modeling framework. By the way, how to seperate the abstract A and a? Is it based on your preset rules, do you have quantitative analysis on the effect of this setting? The paper asserts “synergizes reasoning and control” and “inference-time scaling,” but does not convincingly verify that A constitutes reasoning (vs. correlated labels) or that it explains the gain.
2. **Programmatic labeling may introduce bias or leakage that weakens the “reasoning” interpretation**. High-level actions A_t are produced by a rule-based labeling pipeline from expert trajectories. This creates a risk that A_t encodes privileged structure (e.g., object identity, coordinates, interaction type, or future-dependent cues), making learning easier without reflecting genuine model reasoning. The current writeup does not convincingly rule this out, and there are no targeted stress tests to detect leakage.
3. **Baseline coverage and fairness are not fully convincing given strong SOTA-style claims**.
- The baseline set appears relatively narrow (mostly Qwen2-VL-7B variants plus a few prior agents). In a fast-moving area, “new SOTA” requires broader contextualization and stronger baselines.
- It’s unclear whether all baselines truly use identical preprocessing, relabeling quality, tokenization choices (Text vs Raw action), curriculum steps, and compute in a way that makes comparisons clean.
- Qwen3-MoE is mentioned only in the appendix. if generality is a claim, at least one additional backbone (or a stronger justification) should appear in the main text.
4. **The “end-to-end” framing is somewhat ambiguous or overstated**. “Slow mode” is end-to-end mainly in the sense that one model emits both A and a, but training still uses a two-stage curriculum. “Fast mode” explicitly reintroduces a separate decoder. These are reasonable engineering choices, but they weaken the messaging that the approach fundamentally departs from standard hierarchical pipelines.

## Minor Weakness
 1. **CoA helps and how the fast, slow trade-off should be chosen**
The paper reports a speed–performance trade-off (Table 3), but does not offer:
- a quantitative model of when slow mode is worth the cost,
- an analysis of what kinds of tasks benefit from CoA vs not,
- or an attribution of gains to specific factors (e.g., improved grounding, reduced action ambiguity, better long-horizon consistency).
2. **Generality claims are not fully supported**
- The work is heavily centered on Minecraft only (plus a small transfer to the Language Table). That is still just one domain family; broad claims about general action-space unification would benefit from at least one additional complex domain or a clearer argument that the benchmark stresses the right axes.
- The “generalist OpenHA” evaluation constrains output formats via prompts. It is not clear whether this reflects robust internal generalization or simply learning multiple format-specific behaviors. Stronger tests would involve zero-shot action-space transfer to a new abstraction not seen during training, or explicit cross-format consistency metrics.

---

> ### Author Rebuttal · Authors · 2026-03-31
>
> ### W1. CoA is more than prompt decomposition.
>
> CoA is not a prompting or token-formatting trick. It changes both training and inference: the model is trained to predict a structured intermediate action A together with the primitive action a, rather than primitive actions alone. This is not obtainable from prompting alone, since the base VLM must be fine-tuned on relabeled trajectories to predict and use these abstractions for control.
>
> We have compared against natural-language CoT baseline in Table 12. If CoA's gain mainly came from chain-rule factorization or extra planning tokens, then free-form CoT should provide a similar benefit. However, on Embodied tasks, CoT reaches **27.2** ASR, while CoA reaches **29.7** with MotionCoA and **35.5** with GroundingCoA; on Combat tasks, CoT reaches **22.4**, while CoA reaches **26.3** and **30.6**. This indicates that the gain comes from a structured, action-grounded intermediate representation rather than simply generating extra tokens. Furthermore, CoA is not limited to actions alone; it also incorporates natural language thinking. Unlike CoT, these thoughts are grounded at various levels of action, not just the final primitive action output.
>
> Here, a is the primitive executable action space, while A is a predefined higher-level abstract action space. Our claim is empirical: structured intermediate abstractions improve control more than both direct action prediction and free-form language reasoning.
>
> ---
>
> ### W2. No information leakage.
>
> There is no information leakage here. Our training dataset and evaluation environments are strictly out-of-distribution: no test environments are used during training. Moreover, at inference time, A_t is predicted by the model itself rather than provided by the environment or a human. Thus, the model does not receive privileged information through A_t during evaluation.
>
> ---
>
> ### W3. Baseline coverage is controlled by design.
>
> The purpose of this paper is to isolate the effect of action-space design under a controlled same-backbone, same-domain setting. We therefore fix the base model and control the dataset source, scale, annotation quality, relabeling pipeline, and training setup.
>
> Within this setting, we compare against both prior agents and strong baselines, including ROCKET-1 (CVPR 2025), JARVIS-VLA (ACL 2025), and multiple hierarchical variants with different abstract action spaces. We also compare each action space under its corresponding action format (e.g., text-serialized vs. raw primitive actions), so the conclusions are not tied to one tokenizer or representation choice.
>
> ---
>
> ### W4 / Q2. End-to-end and Fast/Slow mode.
>
> In our formulation, end-to-end means that, in Slow mode, a single autoregressive model predicts both the abstract action A and the primitive action a within one unified sequence model, without a separately learned low-level decoder at inference time. The two-stage curriculum is only for training stability. By contrast, Fast mode reuses a decoupled hierarchical pipeline for efficiency.
>
> In the current submission, the choice between Fast and Slow mode is controlled by the human evaluator for rigor. Empirically, Slow mode is more worthwhile when stronger controllability and capability are needed, while Fast mode is preferable when efficiency matters more. An additional RL follow-up shows that the model can learn more suitable action-space usage autonomously after training, further supporting an empirical speed-capability trade-off.
>
> ---
>
> ### W5. Scope and internal generalization of OpenHA.
>
> Our study is primarily centered on Minecraft. The additional Language Table results provide initial evidence beyond one exact Minecraft setting, but our claim is within-domain breadth rather than universal transfer across all domains.
>
> OpenHA is not a prompt-based method. Its capability is learned through supervised fine-tuning and reinforcement learning on mixed-action data; the system prompt only specifies the output interface during evaluation. When OpenHA is constrained to use the same inference format as a specialist, it still often outperforms that specialist. For example, in Table 4, OpenHA improves over MotionCoA on Combat tasks even when both use the same MotionCoA interface. This supports improved policy quality under matched action interfaces rather than merely learning disconnected format-specific behaviors.
>
> ---
>
> ### Q1. A has causal impact.
>
> We directly test this with intervention in the MotionCoA setting. During inference, instead of using the motion predicted by the model, we replace the intermediate motion with a randomly sampled motion before generating the final low-level action. On Mine tasks, performance drops from **24.8 ± 10.9** to **20.0 ± 3.7**; on Kill tasks, it drops from **25.6 ± 9.8** to **4.0 ± 2.1**. This shows that the intermediate abstract action is not a passive auxiliary label: intervening on A causes predictable changes in final behavior and substantially harms downstream control.

---

### Official Review · Reviewer_fF31 · 2026-03-08

**Soundness:** 2
**Presentation:** 2
**Significance:** 2
**Originality:** 3
**Overall Recommendation:** 4
**Confidence:** 4

**Summary:**

This paper discusses the challenges of defining the abstracted action space for hierarchical LLM agents. Prior benchmarks use domain-specific action tokenizers and have no fair comparison between the abstracted action space. The paper creates a benchmark of 800 Minecraft tasks to evaluate different abstracted action space. The evaluation shows that no action space is the optimal and the performance depends on the task domain. The paper presents a co-training scheme to train with multiple abstract action space. By treating the abstract actions as intermediate reasoning step called chained of actions (CoA), the model can have a fast mode that uses a lightweight policy decoder to decode low-level actions, or a slow mode that use the same LLM to decode low-level actions auto regressively. This design allows training with mix of action spaces based on the sub-datasets. The result showed that the unified agent outperforms the baseline methods and achieves state-of-the-art.

**Compliance With Llm Reviewing Policy:**

Affirmed.

**Final Justification:**

The rebuttal addresses my main concerns. However, the presentation of the paper still needs to improve and the claim about hierarchical agent should be updated to match the real contribution of the paper.

**Key Questions For Authors:**

- When referring to OpenHA’s results in evaluation, are they from the fast or slow mode? It is unclear which mode is used for evaluation and there is no discussion on the results of OpenHA using different modes. Do they perform similarly or have any differences?
- The main idea of co-training is that learning one action space can help another action space. Since the performance have large overlapping std intervals, is it possible to show how one action space relates to another or how one action space contribute to the performance when using another action space in inference? These will be more direct indicator of how co-training can help.
- In Table 2, why OpenHA only reports text action tokenizer? When reporting OpenHA performance, it should be possible to show the results when using different abstracted actions but I cannot find them in the main text and appendix. Also, how do we identify which action space to use in inference given different ones are co-trained?

**Limitations:**

- Limitations of the method and studies are not discussed. It will be helpful to discuss the limitation of the dataset employed for the study and the limitation of the proposed method as listed in the weaknesses/questions.

**Strengths And Weaknesses:**

- Strengths
    - The paper investigates the design of action spaces which was not systematically studied before but has impact on an agent’s performance.
    - The proposed co-training scheme allows generating different abstracted actions and improves success rate across tasks. This suggests that one action representation can potentially help another.

- Weaknesses
    - The set of action spaces does not apply to all task domains. For example, GUI tasks do not use motion primitives, it is unclear the benefit of unifying action spaces as different tasks may need different types of latent actions or semantics of actions.
    - While the experiments show that CoA outperform the previous hierarchical agents, the standard deviation (std) shown in the result are quite large, it is not conclusive that the proposed method outperforms baselines when considering the std intervals.
    - While the agent can use any abstracted actions in inference, the proposed method cannot identify/select one that is suitable for the tasks. Then, action space selection still becomes a deliberate choice in inference.
    - The reference of action space is somewhat confusing. Are the text actions the same as language skill actions? In sec. 2.2, it only defines language skills but later in the experiments only text actions are used. If they are the same, it’s better to be consistent.

---

> ### Author Rebuttal · Authors · 2026-03-31
>
> ### W1. Action spaces across domains.
> We agree that no single action space is universally optimal across domains. This is exactly why we propose CoA: not to enforce one semantic vocabulary, but to unify different abstract action spaces under one framework and shared executable low-level space. We also include results on Language Table, where the action space differs from Minecraft and CoA remains effective.
>
> ---
>
> ### W2. Large standard deviations in open-world evaluation.
> We agree the variance is non-negligible. However, the ±std in Table 2 is computed across tasks within a category, not a confidence interval for a fixed task. Since each category contains tasks with diverse difficulty and dynamics, a relatively large std mainly reflects task heterogeneity. Our conclusions are therefore based on aggregate trends over **800+ tasks** and matched comparisons in Tables 3–4, rather than std overlap alone.
>
> ---
>
> ### W3 / Q3. Action-space selection at inference and why only one OpenHA setting is shown in the main table.
> We agree this should be clarified. In the current submission, OpenHA does not automatically infer the optimal action space during the main evaluation; instead, the output format is specified by the system prompt, which also ensures fair comparison with specialists under matched formats. For readability, the main table reports only one OpenHA setting, namely Text end-to-end, while other prompted formats are reported in Tables 3–4.
>
> Beyond prompt-controlled selection, we also study automatic action-space selection with RL. We choose **10 tasks from each category** (**30 total**), remove the system-prompt restriction, train OpenHA with RL, and evaluate on the same set. Compared with the original OpenHA, the RL-tuned model improves ASR by **+9.9 (+32.9%)** on **MineBlocks**, **+12.6 (+38.8%)** on **KillEntities**, and **+46.9 (+147.0%)** on **CraftItems**. This shows that although action-space selection is prompt-controlled in the main benchmark, RL can enable more autonomous and more effective action-space choice.
>
> ---
>
> ### W4. Clarification on action-space terminology.
> Thank you for pointing this out. `Text Action` denotes primitive environment actions serialized as text, whereas `Language Skill` denotes higher-level abstract actions. The ambiguity arises because both are represented as text tokens for the language model. We will clarify this distinction in the revision.
>
> ---
>
> ### Q1. OpenHA evaluation under Fast vs. Slow modes.
> In Table 2, the reported OpenHA result corresponds to **Text end-to-end (Slow-mode)**, with **30.1 / 32.5 / 31.9 ASR** on **Embodied / GUI / Combat** tasks. Tables 3–4 additionally report OpenHA under prompted CoA-style formats for matched comparison with specialists: under **MotionCoA**, OpenHA reaches **24.8 / 29.5 / 25.6** versus **31.2 / 28.7 / 15.8** for the specialist; under **GroundingCoA**, OpenHA reaches **30.1 / 35.1 / 29.8** versus **31.8 / 30.1 / 28.3**.
> For Fast mode, our added comparison on Minecraft Embodied tasks shows that Hierarchical VLA reaches **28.6 / 39.2 / 30.1** for **Motion / Grounding / Raw**, compared with **27.4 / 37.1 / 27.0** for classical hierarchical agents. We will make the inference mode used for each OpenHA result more explicit in the revision.
>
> ---
>
> ### Q2. Direct evidence for cross-action transfer in co-training.
> We agree that a more direct indicator of cross-action transfer is valuable. To make this effect explicit, we conduct a controlled experiment on the offline dataset: we compare raw-action prediction with and without providing the corresponding high-level grounding action as input. Providing the ground-truth grounding action substantially improves raw-action prediction accuracy: **45.2% vs. 23.7%**, a gain of **21.5 points** and nearly **1.9×** improvement. This directly shows that a higher-level action space can provide useful structure for predicting lower-level actions, so the benefit of co-training is visible not only in final task success but also at the action-prediction level.
>
> ---
>
> ### Limitations.
> Thank you for this helpful suggestion. We agree that the current paper should discuss its limitations more explicitly. In the revision, we will add a dedicated limitations discussion covering both the dataset and the method. On the dataset side, our benchmark is large but still centered on Minecraft, so the main results primarily establish within-domain generalization rather than universal transfer across environments. On the method side, CoA still relies on predefined action spaces, supervised cold-start reasoning traces, and prompt-specified action-space selection at inference time. We view more autonomous emergence of abstractions and automatic inference-time action-space selection as important future directions.

---

> > ### Author Rebuttal · Reviewer_fF31 · 2026-04-04
> >
> > I thank the authors for the additional details and the RL experiments. The RL experiments clarify my main concern of manually choosing an action space. The clarification of action space definition and the limitations will help understand the merit of the paper. And I agree with reviewer Bc9X that the choice of action space is a shallow hierarchy, and would suggest the authors to update the paper accordingly.

---

> > > ### Author Response · Authors · 2026-04-07
> > >
> > > Thank you for acknowledging our rebuttal. We especially appreciate your suggestion regarding the RL experiments — it directly motivated us to add the new analysis, which we believe has meaningfully strengthened the paper. Your observation about the action space definition was also very helpful and led us to rethink how we present these concepts.
> > >
> > > We fully agree with the suggestion regarding the hierarchy framing. In the revision, we will clearly distinguish it from classical hierarchical RL. We believe this more precise expression better reflects the actual contribution.

---

### Official Review · Reviewer_Bc9X · 2026-03-12

**Soundness:** 3
**Presentation:** 2
**Significance:** 2
**Originality:** 2
**Overall Recommendation:** 3
**Confidence:** 3

**Summary:**

This paper studies an important question for agent design: what action space should a generalist agent use? The authors argue that no single action space is best across all tasks, and they support this through a large study in Minecraft. To address this they propose Chain-of-Action, where the model first predicts an intermediate abstract action and then generates the low-level executable action. They also train an all-in-one agent across multiple action spaces and show improved performance over several baselines. The paper is ambitious, the experimental scope is large, and the benchmark release could be useful to the community.

**Compliance With Llm Reviewing Policy:**

Affirmed.

**Key Questions For Authors:**

1. If possible, Tone down the hierarchy claim or define more clearly what kind of hierarchy this is? now the work reads more like a action-generation framework than a strong new hierarchical agent design.

2, Would it be fine to add an ablation that isolates the source of gains. For example, compare: primitive action only, abstract action only abstract + primitive without sequential conditioning? This would clarify whether the benefit really comes from hierarchical decomposition.

3. Perhaps try discussing the difference from classical hierarchical RL more explicitly?

**Limitations:**

I am not yet convinced that the main conceptual contribution is as strong as the paper presents it. The proposed “hierarchy” feels relatively shallow, and much of the benefit may come from better action representation rather than a genuinely new hierarchical reasoning mechanism. The results are interesting.

**Strengths And Weaknesses:**

### Strenghts
1. The paper asks a good question.
The action-space question is central to agent design, and the paper is right that it is often under-discussed. The finding that different action spaces work better for different task types is useful.

2. The empirical effort is strong.
The paper includes a large number of tasks, multiple action-space variants, several baselines, and a broad evaluation setup. This is a serious experimental study, and I appreciate the scale.

3. The benchmark and release are valuable.
Releasing tasks, datasets, code, and checkpoints could make this work useful even beyond the paper itself.

4. The all-in-one training result is interesting.
Even if I am not fully convinced by the hierarchy framing, the idea of mixing action spaces during training and learning a more robust policy is promising.

### Weakness


1. The "hierarchy" feels shallow.
This is my main concern. In classic hierarchical control, the high-level policy sets a subgoal or option, and the low-level policy executes over a meaningful temporal span. Here, the hierarchy mostly consists of generating an intermediate abstract action followed by a primitive action. That does add structure, but it is closer to action tokenization with an extra reasoning step than to a deep hierarchical planner. The paper would be stronger if it positioned this more modestly. A cleaner way to say it is:
   - The hierarchical structure consists of generating an intermediate abstract action, treated as a reasoning step, followed by low-level environment actions. However, the hierarchy is relatively shallow and mainly operates through action formatting rather than explicit multi-level planning.

2. The gains do not feel large enough to fully justify framing.
paper claims a new state of the art, but the improvements over strong baselines do not always feel substantial enough to support the bigger conceptual claim. Given how strongly the paper emphasizes a new hierarchy-based framework, I expected either clearly stronger gains or stronger evidence that the hierarchy itself is the reason for those gains.

3. Generalization may  not be fully convincing.
The benchmark has 800+ tasks, which sounds large and impressive as well, but task count alone is not the same as generalization. The evaluations are still within Minecraft, and the notion of generalization mostly comes from new seeds and task variations inside the same environment. The paper should be more careful not to overstate this point or be transparent about it.

4. It is not always clear whether CoA helps because of hierarchy or because of extra supervision.
Since the model gets both abstract and primitive action formats, part of the gain may simply come from richer supervision or a better output structure. I do not think the current ablations cleanly separate these explanations.

---

> ### Author Rebuttal · Authors · 2026-03-31
>
> ### W1 / Q1 / Q3: What kind of “hierarchy” is CoA, and how is it different from classical hierarchical RL?
>
> We agree that this distinction should be stated more precisely. CoA is **not** intended as a classical deep hierarchical planner with long-horizon options or multiple learned planning levels. Instead, it uses a **two-level abstract-to-primitive action decomposition**, where the model first predicts an abstract action and then low-level environment actions. In this sense, the “hierarchy” in CoA is explicit but relatively shallow.
>
> Importantly, our framework supports two inference modes. In **Fast mode**, it is closer to a conventional hierarchical agent: the high-level model predicts an abstract action, and a lightweight decoder generates primitive actions. In **Slow mode**, the same decomposition is realized inside a single autoregressive model as sequential abstract-action then primitive-action prediction. Our goal is therefore not to introduce a new deep hierarchical RL algorithm, but to show how hierarchical abstraction and end-to-end VLA can be unified within one framework.
>
> We will revise the paper to tone down the hierarchy claim and clarify more explicitly that CoA should be viewed as a bridge between hierarchical agents and VLA models, rather than as a direct replacement for classical hierarchical RL or option-based control.
>
> ---
>
> ### W2 / W4 / Q2: What is the real source of gains in CoA?
>
> We agree that this is the central question, and our evidence suggests that the gains come from both (i) richer supervision from multiple action spaces and (ii) the sequential CoA decomposition itself.
>
> First, regarding richer supervision from multiple action spaces, when using the same Fast-mode inference pipeline and the same lightweight decoder as classical hierarchical agents, Hierarchical VLA with Fast mode still improves over classical HA on Minecraft Embodied tasks:
>
> | Action Format / Space | Classical Hierarchical Agent | Hierarchical VLA with Fast Mode |
> |---|---:|---:|
> | Motion | 27.4 | 28.6 |
> | Grounding | 37.1 | 39.2 |
> | Raw | 27.0 | 30.1 |
>
> Second, to more directly isolate the source of gains, we add the following ablation:
>
> | Setting | Embodied Task (Grounding) | Combat Task (Motion) |
> |---|---:|---:|
> | Primitive Action Only | 27.0 | 10.0 |
> | Abstract Action Only | 27.4 | 4.3 |
> | Abstract + Primitive without Sequence | 28.6 | 7.9 |
> | Abstract + Primitive with Sequence (CoA) | 31.2 | 15.8 |
>
> Combining abstract and primitive actions without sequential conditioning already improves over using either one alone, indicating that richer supervision helps. However, the full CoA sequential formulation gives the strongest results, improving from **28.6 to 31.2** on the Embodied task and from **7.9 to 15.8** on the Combat task. This indicates that the benefit is not only from extra supervision or output structure, but also from the explicit sequential abstract-to-primitive decomposition.
>
> We will revise the paper to make this causal interpretation of the gains more explicit.
>
> ---
>
> ### W2: Magnitude of improvements and justification of the framework
>
> We respectfully clarify that our contribution is not only higher raw scores, but the CoA formulation itself, which unifies the benefits of different action spaces within one framework. A more controlled way to evaluate this is to compare CoA against prior hierarchical agents under the same action space. In Table 3, this already shows clear gains: for Motion actions, CoA improves ASR from **0.0 to 28.7** on GUI tasks and from **4.3 to 15.8** on Combat tasks; for Grounding actions, CoA improves ASR from **6.7 to 30.1** on GUI tasks and from **26.5 to 28.3** on Combat tasks. These controlled comparisons indicate that the gains are not only from changing the action space, but from the CoA formulation itself.
>
> Beyond this, the all-in-one OpenHA model trained on the full CoA data is the strongest Minecraft agent in our benchmark in Table 2.
>
> Regarding the large variance, we agree that open-world Minecraft is inherently high-variance due to diverse seeds, initial states, and task dynamics. This is exactly why we scale the evaluation up to 800+ tasks, so that conclusions are drawn from broad aggregate trends rather than a small number of favorable runs.
>
> ---
>
> ### W3: Generalization beyond Minecraft
>
> We agree that the current empirical coverage should be stated more carefully. In the present submission, the main benchmark demonstrates within-Minecraft generalization across a broad set of 800+ tasks, rather than unrestricted transfer to arbitrary environments.
>
> Beyond Minecraft, we also include additional tabletop manipulation experiments on the Language Table benchmark in the main paper (Table 5), where OpenHA again outperforms single-action agents. We view these results as preliminary evidence that the benefits of CoA and multi-action training can transfer beyond Minecraft, but not as a definitive claim of universal generalization.

---

> > ### Author Rebuttal · Reviewer_Bc9X · 2026-04-04
> >
> > Thank you to the authors for the detailed responses and additional analysis. I appreciate the effort put into the rebuttal. While some points were clarified, some aspects remain insufficiently convincing to me, so I will keep my current score.

---

> > > ### Author Response · Authors · 2026-04-07
> > >
> > > Thank you for taking the time to read our rebuttal and for acknowledging the additional analysis we provided. We genuinely appreciate your engagement throughout this process.
> > >
> > > We understand that some aspects remain insufficiently convincing to you, and we would very much like to address them. However, since your acknowledgement did not specify which particular concerns are still unresolved, we are not sure where to focus our efforts. We want to make sure we do not miss anything important.
> > >
> > > Could you kindly clarify which specific aspects you still find unconvincing? We are happy to provide further clarification on any of these points. The discussion period is a valuable opportunity for us to improve the work, and we would appreciate any specific feedback you can share.
> > >
> > > Thank you again for your careful review!

---

### Official Review · Reviewer_6REB · 2026-03-13

**Soundness:** 4
**Presentation:** 3
**Significance:** 3
**Originality:** 3
**Overall Recommendation:** 5
**Confidence:** 2

**Summary:**

The authors introduce OpenHA, open-sourced hierarchical agent models with a benchmark of 800 tasks in Minecraft. These hierarchical agents considered all contain LLMs and vision encoder that takes in text-labeled trajectories + visual embeddings. The authors introduce Chain of Actions, where generated abstract actions condition low level actions, and both the ``reasoning step'' and low level action output are trained together on offline datasets: the trajectory data consists of triplets of abstract action, low level actions, observations, and the overall text instructions. Then, authors run experiments in Minecraft that (1) show different abstracted action spaces have very different performance on different benchmarks, (2) reasoning action-latent model (where reasoning and action generation done end-to-end) achieves higher performance relative to just training abstract model + separate, cheaper policy decoder, and (3) mixing abstract action spaces leads to better performance than any one individual abstract action space.

**Compliance With Llm Reviewing Policy:**

Affirmed.

**Final Justification:**

Fully addressed my questions in the rebuttal. The RL post-training results are a nice addition, and should be centered + expanded upon in the paper. Open access is encouraging, though I haven't looked carefully at the codebase.

Generally a well-written and well-organized paper. I agree it's a bit different from other heirarchical RL methods, e.g. options-based methods. But it's still clear to me that the method is hierarchical in nature and, if the unsupervised latent actions learning is more fleshed out, lays a convincing path to do hierarchical RL w/ semi-supervised method.

I am also of the opinion Minecraft is a pretty tough, partially-observed benchmark, so am personally not concerned on that front.

Thus I keep my 5 after taking into account all reviews + rebuttals, with the caveat that this is out of my domain so have low confidence.

**Key Questions For Authors:**

See weaknesses:

- Reasoning, here, is a supervised problem. Can reasoning traces be learned in an end-to-end fashion? E.g., where the abstract action space itself is emergent?

- What do the actual abstract actions look like @ test time for OpenHA when mixing all of the abstract actions during training?

**Limitations:**

yes

**Strengths And Weaknesses:**

Strengths:

Overall, this is a well-written paper that is significant (1) for the contribution to the RL community w/ open-access hierarchical VLA models, (2) answering basic questions about the utility of different action latents from recent SOTA works, and (3) introducing a framework to unify arbitrary action latents, framed using CoT (``CoA'').

- Grounding in prior work is thorough. Differences between VLA, hierarchical agent model, hierarchical VLA for the Minecraft tasks and paper's contributions made clear from the start of the paper + abstract.

- I clicked on the anonymous GitHub and do not see anything anything yet. But, assuming that the authors will eventually ``releasing the OpenHA (Open HierarchicalAgents) suite, which includes our benchmark of over 800 tasks, curated datasets, source code, and all model checkpoints,'' this seems like a very useful contribution to the broader RL community if the code is readable, well-formatted, and maintained.

- Three hypotheses explored in experiments unifies existing VLA work with various action spaces, and demonstrates that one can effectively incorporate arbitrary abstractions into one policy.

- Experimental setups are clear: e.g., datasets used, action abstractions, additional implementation details in the appendix, etc.

Weaknesses:

Mostly minor clarifications and questions about the CoA framing, especially in relation to CoT.

- Stating Fast mode of Hierarchical VLA and prior Hierarchical Agent Model are similar/the same earlier in the paper. I see it's mentioned in the Experiments, but was confused at first about the distinction.

- Here, reasoning is a supervised problem -- the reasoning traces/thoughts themselves are included in the dataset. While beneficial re: Q2 /T3 in the experiments, I'm curious about learning multi-step reasoning traces in an unsupervised/complete end-to-end fashion where the start is the labeled abstraction (whether grounding, motion, or skill discussed in the appendix).

- Adding onto ^^, what do the actual abstract actions look like @ test time for OpenHA/mixing all of the abstract actions during training? I couldn't find it after a quick scroll through of the appendix, but I'm curious whether OpenHA automatically outputs the abstract action in the task-optimal action space re: Q1, or whether the abstract action ends up mixing the various forms.

---

> ### Author Rebuttal · Authors · 2026-03-31
>
> ### W1: Content in the anonymous link is not accessible.
>
> Thank you for pointing this out. The previous anonymous repository link had expired, which is why the content was not accessible. We have refreshed the anonymous repository referenced in the paper. As described in the appendix, the release includes the codebase and evaluation pipeline, and we will ensure that the full suite of datasets and model checkpoints is made available through stable public hosting at release time. We agree that the practical value of OpenHA depends on the release being readable, well-organized, and reproducible, and we will make that a priority.
>
> ---
>
> ### W2 / Q1: Difference between Fast mode of Hierarchical VLA and prior Hierarchical Agent Model is not clear.
>
> We agree that this distinction should be introduced earlier and more explicitly. The Fast mode of Hierarchical VLA is architecturally similar to a prior hierarchical agent at inference time, in the sense that both first produce an abstract action and then rely on a lightweight decoder to generate primitive environment actions. However, they differ in training: in a standard hierarchical agent, the high-level policy and low-level policy are trained separately, whereas our Hierarchical VLA is trained with the CoA objective to jointly model abstract actions and low-level actions within one unified framework.
>
> We additionally include a direct comparison under Fast mode on Minecraft Embodied tasks:
>
> | Action Format / Space | Classical Hierarchical Agent | Hierarchical VLA with Fast Mode |
> |---|---:|---:|
> | Motion | 27.4 | 28.6 |
> | Grounding | 37.1 | 39.2 |
> | Raw | 27.0 | 30.1 |
>
> These results show that, although the fast inference path resembles a hierarchical agent structurally, the learned policy is still different due to the unified CoA training. We will revise the paper to clarify this distinction in the method section rather than only in the experimental discussion.
>
> ---
>
> ### W3: Unsupervised learning of reasoning traces is not clear.
>
> This is an excellent point. In the current paper, the reasoning traces are indeed supervised and mainly serve as a cold-start signal that stabilizes learning. Importantly, these traces are not manually written; they are produced by an automatic labeling pipeline built on top of the underlying trajectories. More broadly, the CoA formulation does not require reasoning traces to remain fully supervised forever. A natural next step is to optimize the reasoning process end-to-end from task outcomes, for example, through reward-based post-training, so that the model can refine how it composes multi-step abstract actions beyond the initial supervised signal.
>
> We further validate this with RL post-training. Specifically, we select 10 tasks from each category (30 tasks in total), remove the system-prompt restriction so that OpenHA can freely choose its abstract actions, train it with RL, and evaluate on the same task set. Compared with OpenHA, the RL-tuned OpenHA improves ASR by **+9.9** (**+32.9%**) on MineBlocks, **+12.6** (**+38.8%**) on KillEntities, and **+46.9** (**+147.0%**) on CraftItems. This suggests that the model can go beyond the initial supervised traces and learn more effective multi-step reasoning behaviors from downstream rewards, including the emergence of abstract actions not seen during the initial supervised stage. We will clarify in the revision that the present submission uses supervised traces as initialization, while RL-based refinement of more autonomous reasoning is a promising extension.
>
> ---
>
> ### W4 / Q2: Abstract actions at test time are not clear.
>
> We provide concrete rollout examples in Appendix A and Fig. 1, which show both the abstract actions and corresponding low-level environment actions.
>
> In OpenHA, we use a system prompt to explicitly specify the allowed abstract action format. Without this constraint, the model can generate mixed forms and implicitly choose among action spaces, though not always optimally.
>
> As shown in W3, our further RL experiments also enable the model to learn to select more effective action spaces autonomously.

---

> > ### Author Rebuttal · Reviewer_6REB · 2026-04-01
> >
> > Thank you for your thorough response. The additional experiments and explanations fully addressed my questions about the work, and I maintain a strong support of acceptance.
> >
> > Overall, the new RL post-training experiments are a clear technical contribution which, in conjunction with a thorough benchmarking, evaluation, and open-sourcing of hierarchical RL models in one of the most challenging, embodied benchmarks is significant.
> >
> > The link to the repository does not work for me (https://anonymous.
> > 4open.science/anonymize/OpenHA-ACFE), but authors are not allowed to share additional links beyond what already exists with text. It may be an issue on my side. If you can get the same link working, **that would raise my confidence in my rating which hinges on assuming that the codebase is easy-to-parse, as thorough as the paper claims, and accessible, but it would not change my score**. Currently, my confidence is a 2 because I have not seen the code.
> >
> > -----
> >
> > > W1
> >
> > The openness of the work satisfies the condition of Significance. To my knowledge, there are no/few workable open-source Minecraft for HRL repositories -- this benchmark is a clear step up from the canonical MuJoCo or Atari-type environments in evaluating tasks like reasoning, and is more general/embodied than challenging combinatorial tasks.
> >
> > > W2/Q1
> >
> > Makes sense, thank you for the clarification. This difference should be made more salient and sooner in the paper.
> >
> > > W3
> >
> > This is strong, and looks much more like CoA as true CoT. Including and highlighting some of these results in the main body of the paper text, in my view, would increase the significance and originality of this work -- e.g. in the abstract and the introduction.
> >
> > > W4/Q2
> >
> > Thanks for the clarification.

---

> > > ### Author Response · Authors · 2026-04-02
> > >
> > > Thank you for your positive response. The repository link requires updating "anonymize" in "anonymize/OpenHA-ACFE" to "r". Due to policy restrictions, we are unable to include the latest training and RL experiment code in the repository. Therefore, only the Agent's structure, inference, and task code are available (unchanged from the paper submission).
> > >
> > > We promise that the **benchmarks, training and inference code, training datasets, and model weights** mentioned in the paper will be included in the updated version.
> > >
> > > Thank you again for your response.

---

### Decision · Program_Chairs · 2026-04-30

**Decision:**

Accept (regular)

**Comment:**

We have carefully considered all the available evidence (the reviews, author responses, discussions, and the paper).

The paper is sufficiently sound and significant, and of broad interest to the ML community. It proposes and evaluates OpenHA (Open Hierarchical Agents) suite, based on a Chain-of-Action (CoA) novel framework that integrates  high-level abstracted actions and low-level control actions.

A weakness is that the repository link included in the paper (https://anonymous.4open.science/anonymize/OpenHA-ACFE) could not be accessed during the reviewing and rebuttal process.

We believe that the combined technical and empirical strengths and contributions, and the relevance to ICML, recommend the paper for acceptance.

We encourage the authors to carefully consider and address the reviewers' comments and suggestions, and - as mentioned in their responses to reviewers, also include benchmarks, training and inference code, training datasets, and model weights in the final version of the paper.